# TIMESERIESGYM: A SCALABLE BENCHMARK FOR (TIME SERIES) MACHINE LEARNING ENGINEERING AGENTS

## ABSTRACT

We introduce `TimeSeriesGym`, a scalable benchmarking framework for evaluating Artificial Intelligence (AI) agents on time series machine learning engineering challenges. Existing benchmarks lack scalability, focus narrowly on model building in well-defined settings, and evaluate only a limited set of research artifacts (e.g., CSV submission files). To make AI agent benchmarking more relevant to the practice of machine learning engineering, our framework scales along two critical dimensions. First, recognizing that effective ML engineering requires a range of diverse skills, `TimeSeriesGym` incorporates challenges from diverse sources spanning multiple domains and tasks. We design challenges to evaluate both isolated capabilities (including data handling, understanding research repositories, and code translation) and their combinations, and rather than addressing each challenge independently, we develop tools that support designing multiple challenges at scale. Second, we implement evaluation mechanisms for multiple research artifacts, including submission files, code, and models, using precise numeric measures and *optionally* LLM-based qualitative assessments. This strategy complements objective evaluation with subjective assessment when appropriate. Although our initial focus is on time series applications, our framework can be readily extended to other data modalities, broadly enhancing the comprehensiveness and practical utility of agentic AI evaluation. We open-source[1] our benchmarking framework to facilitate future research on the ML engineering capabilities of AI agents.

## 1 INTRODUCTION

AI agents (9; 11) have shown growing promise in automating machine learning (ML) and data science (DS) workflows. Fueled by large language models (LLMs), they can reason about context, adapt to diverse tasks, and iteratively refine solutions over long horizons. Such capabilities have the potential to significantly reduce the mundane, mostly manual efforts in ML engineering and improve the overall productivity of ML practice. To measure progress in this area, several benchmarks (9; 3; 12; 17; 22; 1; 20; 18) have been proposed to evaluate AI agents on ML and DS tasks.

However, existing benchmarks have important limitations. First, many of them source ML challenges primarily from popular competitions such as Kaggle, which are well-structured and do not fully capture the complexity of real-world ML tasks. Second, evaluations are typically outcome-based, focusing on overall task completion or eventual model performance metrics such as accuracy, while combining and obfuscating the impact of multiple component skills that jointly determine the outcomes, such as effective data wrangling or code quality improvement capabilities. Third, current benchmarks lack scalability, as tasks have to be manually curated and cannot be developed at scale.

To enable efficient evaluation of AI agents in realistic ML scenarios, we propose `TimeSeriesGym`, a scalable and agent-agnostic benchmarking framework for evaluating AI agents on time series ML engineering tasks. `TimeSeriesGym` currently consists of 33 challenges that span 8 unique time series problems (forecasting, classification, time series understanding), from more than 15 domains (healthcare, finance, epidemiology). Our benchmark covers both Kaggle-style challenges

---

[1] https://anonymous.4open.science/r/TimeSeriesGym-9CF6/

Figure 1: `TimeSeriesGym` is a scalable benchmarking environment for ML engineering agents. It currently features 33 time series challenges across 8 unique time series problems, spanning more than 15 domains. Challenges are either carefully designed based on real-world ML practice, or sourced from Kaggle competitions and GitHub repositories. `TimeSeriesGym` includes key mechanisms to enable efficient and scalable generation of new challenges. Our evaluation methodology complements precise quantitative metrics with *optional* qualitative assessment, and provides specialized tools to grade various artifacts generated during ML engineering. `TimeSeriesGym` is compatible with many different agent types, even those with fundamentally distinct designs.

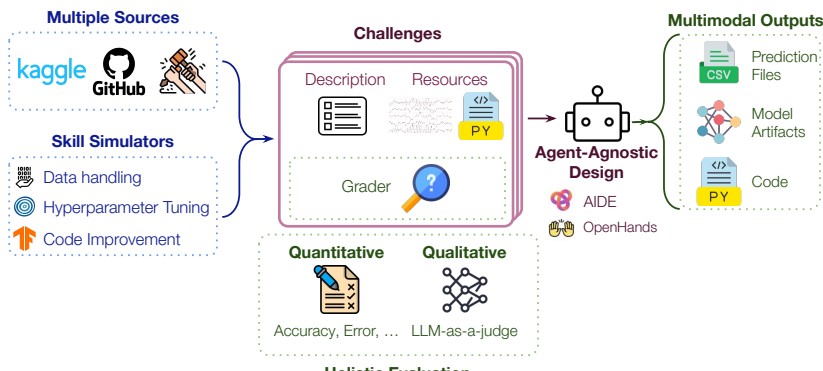

and original tasks carefully designed based on real-world ML engineering practice. While we focus on time series analytics due to its prevalence in applications and under-representation in existing agent benchmarks, our framework is modality-agnostic in principle and can be easily extended to handling other data modalities (e.g., images, text, audio) via the accompanying tools for scalable task generation. `TimeSeriesGym` provides an interactive `gym` environment compatible with various types of agent scaffolding, allowing seamless evaluation of agents of different types and collection of their trajectories. Beyond benchmarking, this allows `TimeSeriesGym` to also serve as a data flywheel for future agent improvement through post-training using the collected trajectory data.

Our contributions are as follows:

1. We propose `TimeSeriesGym`, the first open source benchmark for AI agents on time series ML engineering tasks (Fig. 1). Beyond benchmarking, `TimeSeries`"Gym" can be easily used as a data flywheel to post-train the next generation of ML engineering agents.

2. `TimeSeriesGym` offers a scalable task generation mechanism that reduces manual efforts in task curation and ensures long-term sustainability of the benchmark.

3. `TimeSeriesGym` provides a comprehensive framework that evaluates **multimodal agent outputs** (e.g., prediction files, models, code) across **specific ML skills** (e.g., data handling, model improvement), using a **holistic approach** that complements quantitative metrics (e.g., accuracy) with *optional* qualitative assessment (e.g., LLM-as-a-judge for code utility).

## 2 RELATED WORK

**Machine learning agent benchmarks.** Several benchmarks have been proposed to evaluate LLM agents on automating ML and DS tasks. These benchmarks are typically structured around three key components: (1) task curation, (2) agent capabilities being evaluated, and (3) evaluation protocol. Benchmarks differ in how they curate ML/DS tasks. For example, MLE-bench (3) and DSBench (12) compile tasks from online competition platforms such as Kaggle, while other benchmarks source tasks from ML-related GitHub repositories (1; 20) or hand-craft tasks based on ML research problems or engineering workflows (9; 17; 22; 18). With regard to agent capabilities, some benchmarks (3; 9; 17; 22; 12) focus on comprehensive ML science skills by evaluating agents on end-to-end problem solving skills, while others (1; 20; 18) focus on more modular engineering-oriented capabilities within the ML pipeline, such as using GitHub repositories or integrating APIs. Evaluation protocols also differ in output formats and granularity. MLE-bench (3) and DSBench (12) require agents to output

Table 1: Comparison of `TimeSeriesGym` with existing ML/DS agent benchmarks. Categories include **Number:** total and time series (TS) tasks in each benchmark, where each task corresponds to a unique data source (e.g., a single Kaggle competition or GitHub repository); **Source:** task origins (K: Kaggle, G: GitHub, H: Hand-crafted); **ML Capability:** coverage of ML **science** tasks (e.g., modeling, open-ended research) and **engineering** tasks (e.g., repository utilization, API integration); and **Evaluation:** capabilities for evaluating **multimodal** outputs (e.g., prediction files, model artifacts), specific ML **skills** (e.g., data handling, model improvement), and from a **holistic** perspective (including both quantitative metrics (e.g., accuracy, mean absolute error) and qualitative evaluation (e.g., code review)). We use "+" to indicate `TimeSeriesGym`'s scalability which enables the generation of an unlimited number of new challenges using the tools provided.

| | Number | | Source | | | ML Capability | | Evaluation | | |
|---|---|---|---|---|---|---|---|---|---|---|
| | Total | TS | K | G | H | Science | Engineering | Multimodal | Skill-based | Holistic |
| MLE-bench (3) | 75 | 3 | ✓ | ✗ | ✗ | ✓ | ✗ | ✗ | ✗ | ✗ |
| MLAgentBench (9) | 13 | 1 | ✓ | ✗ | ✓ | ✓ | ✗ | ✓ | ✗ | ✗ |
| MLGym (17) | 13 | 0 | ✓ | ✗ | ✓ | ✓ | ✗ | ✓ | ✗ | ✗ |
| RE-Bench (22) | 7 | 0 | ✗ | ✗ | ✓ | ✓ | ✗ | ✓ | ✗ | ✗ |
| DSBench[2] (12) | 74 | 5 | ✓ | ✗ | ✗ | ✓ | ✗ | ✗ | ✗ | ✗ |
| SUPER[3] (1) | 45 | 0 | ✗ | ✓ | ✗ | ✗ | ✓ | ✗ | ✗ | ✓ |
| ML-Bench (20) | 18 | 1 | ✗ | ✓ | ✗ | ✗ | ✓ | ✗ | ✗ | ✗ |
| ML-Dev-Bench (18) | 30 | 0 | ✗ | ✗ | ✓ | ✗ | ✓ | ✓ | ✓ | ✗ |
| TimeSeriesGym (Ours) | 23+ | 23+ | ✓ | ✓ | ✓ | ✓ | ✓ | ✓ | ✓ | ✓ |

results in specific formats (e.g., CSV files) that can be directly scored using predefined metrics such as accuracy, while other benchmarks (9; 17; 22) allow for more flexible outputs in addition to prediction files, such as model artifacts and code. ML-Dev-Bench (18) further extends the evaluation by specific skills (e.g., data handling, model improvement), while SUPER (1) provides a more holistic evaluation by combining outcome-based evaluation with qualitative code inspection to assess agents' progress towards completing the tasks. `TimeSeriesGym` is most similar to MLE-Bench in sourcing Kaggle competitions, but uniquely emphasizes time series modeling (an underrepresented modality), includes original ML engineering challenges beyond competitions, and provides granular skill assessment, and holistic holistic multi-artifact evaluation (see difference in Sec. D).

**Scalable dynamic benchmarks and holistic evaluation.** Scalable benchmarks reduce manual data curation efforts by generating target problems at scale using carefully designed templates (2; 23) or data engines (8), among which TimeSeriesExam (2) further improves problem sample quality by applying Item Response Theory (IRT) (4; 7) to intelligently select questions with contextualized difficulty and appropriate discrimination. To remain effective against data contamination from LLM pretraining, dynamic benchmarks such as GAIA (16) and LiveCodeBench (10) propose to continually incorporate problems newly released after LLM training cut-offs. While most benchmarks target specific capabilities, holistic evaluation (13; 5) provides a comprehensive picture through evaluating models on a wide range of datasets and tasks across diverse domains using multiple complementary metrics, to capture both the breadth and depth of model capabilities.

## 3 TIMESERIESGYM

`TimeSeriesGym` is envisioned as a scalable benchmarking environment for time series machine learning engineering. The current version features 33 challenges from 23 unique data sources across 8 unique time series problems, spanning more than 15 domains. These challenges evaluate AI agents on a range of realistic ML engineering skills beyond just model development, including data labeling, model selection, and the utilization, improvement, and migration of research code (Tab. 4). `TimeSeriesGym` also provides tools for rapidly developing new challenges to test specific skills and for evaluating the diverse artifacts commonly generated during ML engineering processes.

Each challenge in TimeSeriesGym is organized with a consistent structure: (1) **resources** including datasets, code repositories, related paper(s) and documentation relevant to the challenge; (2) a

---

[1]For DSBench, we include only data modeling tasks, while excluding data analysis tasks as they are not directly relevant to our work.

[2]For SUPER, we include repositories used to create the Expert and Masked sets of the benchmark.

**description file** that outlines the challenge parameters, available resources, and provides specific instructions and hints for successful completion; and (3) **challenge-specific grading functions** to evaluate agent submissions. Some challenges also include leaderboards to rank agent submissions against human performance. These leaderboards are readily available for, e.g., challenges derived from Kaggle competitions.

**The challenges in TimeSeriesGym** are derived from Kaggle competitions (currently, n = 13) and popular benchmarks and research code repositories for time series modeling (`TimeSeriesGym Originals`, n = 14). We prioritized challenges that reflect core skills that are regularly exercised by ML engineers, researchers, and data scientists.

Each challenge is specifically chosen or designed to evaluate one or more of the following skills: (1) *Data Handling:* Ability to handle missing data, use data labeling tools, and utilize multi-source data for model building. (2) *Modeling:* Ability to develop useful time-series ML models, tune hyperparameters, perform model selection, and understand, utilize, migrate and improve the quality of research code. (3) *Benchmarking:* Training and rigorously evaluating ML models using standard benchmarks. In selecting these challenges, we aimed at a broad coverage across diverse domains (e.g., healthcare, finance, epidemiology) and time series problems (forecasting, classification, time series understanding). Tab. 4 provides a comprehensive overview of each challenge within `TimeSeriesGym`, including its domain, core problem, evaluation metric, and the skills required to address it.

To identify Kaggle challenges for inclusion in `TimeSeriesGym`, we began with the Meta Kaggle dataset (19), focusing specifically on Featured and Research competitions. Featured competitions are real-world ML challenges that pose difficult, commercially oriented prediction problems, while Research competitions focus on problems that may not have clean or straightforward solutions[4]. We employed `Gemini 2.0 Flash` to analyze competition descriptions and titles, identifying 453 competitions that likely involve time series data. Subsequently, we ranked these competitions by participant count, maximum reward, and presence of a public leaderboard. Then from top 100 ranked high-quality competitions (see Tab. 8), we manually selected a diverse set of competitions to ensure comprehensive coverage across problem type (e.g., forecasting, classification), domain (e.g, finance, healthcare), and research or engineering complexity (e.g., organization of datasets), while also requiring public data availability, and a permissive license.

To complement the selected Kaggle challenges, we include 14 `TimeSeriesGym` Original challenges, manually curated from existing open-source time series datasets and GitHub repositories based on recommendations from experienced ML engineers and researchers (see Tab. 5 for the original and modified source codes and licenses). These challenges are specifically designed to evaluate advanced skills that Kaggle competitions typically cannot easily assess, yet are essential for effective ML engineering. Examples include utilizing state-of-the-art models (e.g., `Implement the MOMENT (6) time series foundation model for anomaly detection`), migrating frameworks (e.g., `Convert ResNet-1D classification models from TensorFlow to PyTorch`), and improving research code quality (e.g., `Improve PTB-XL ECG Classification Code`). These capabilities represent critical competencies of skilled ML engineers that extend beyond the scope of standard Kaggle-like competitions.

Running experiments on `TimeSeriesGym` can be resource-intensive and costly. Therefore, we propose `TimeSeriesGym-Lite`, a carefully selected subset of six challenges designed to efficiently evaluate AI agents on critical ML engineering skills while maintaining coverage across multiple domains and time series problems, with statistically similar difficulty to `TimeSeriesGym` (Sec. B). This collection enables rapid and cost-effective assessment of novel AI agents without sacrificing the diversity of skills being tested (see Tab. 6).

**Multimodal, skill-based, holistic evaluation.** Existing benchmarks typically summarize agent performance using metrics such as accuracy, completion rate, or competition rankings (3). Although these metrics provide useful summaries, they do not offer much actionable feedback for improvement. `TimeSeriesGym` addresses this limitation through an evaluation framework designed to provide specific actionable feedback through multiple complementary approaches. First, we design challenges that isolate and test specific skills, such as handling missing data (e.g., `Optiver Realized Volatility Prediction with Missing Data`). Poor performance on such challenges

---

[4]`https://www.kaggle.com/docs/competitions`

clearly indicates potential skill gaps, enabling developers to focus their efforts on specific skills. Second, we develop fine-grained evaluation tools that assess multiple dimensions of performance simultaneously. For example, in code migration tasks (e.g., `Convert ResNet from TensorFlow to PyTorch`), our evaluation tools examine whether an agent follows instructions and naming conventions, completes all required function definitions, in addition to successful execution– providing a multidimensional performance profile rather than a binary success/failure indicator.

Our evaluation methodology deliberately combines multiple assessment approaches: quantitative metrics (accuracy, mean absolute error), programmatic analysis (regular expression matching, code inspection), and optional qualitative evaluation (LLM-as-a-judge) (see Appendix G). Each challenge in `TimeSeriesGym` is evaluated using quantitative metrics (based on rules) and optionally subjective metrics (LLM judge). Although LLM-based evaluation offers valuable insight, especially for open-ended tasks such as research code enhancement, we recognize that LLMs can be inconsistent and prone to hallucination. Therefore, we primarily rely on quantitative metrics and strategically complement them with subjective assessments, creating a holistic evaluation system that leverages the strengths of both approaches. This hybrid approach mimics code reviews in software engineering practice, which includes both tests based on static analysis and human code reviews.

Furthermore, `TimeSeriesGym` provides specialized tools to grade diverse artifacts generated throughout the ML engineering life cycle– from submission files (`CSV`, `H5`, etc.) to source code (`.py`) and trained models (`.pth`, `.pkl`)– enabling comprehensive assessment of the entire development process rather than focusing solely on final outputs.

**Generating challenges at scale.** We provide key mechanisms that enable scalable generation of new challenges. Here, *scalability* refers not only to adding new challenges efficiently, but more importantly to the *flexibility* to support multiple design choices across various components of the benchmark. By design, `TimeSeriesGym` scales along several key dimensions: (1) **Skill-specific competitions:** We provide specialized tools (e.g., missing data simulator) that can be paired with any "base" competition to create a large variety of targeted, skill-specific competitions. (2) **Agent outputs:** Our grading tools support the evaluation of diverse agent outputs, including prediction files, model artifacts, and code, allowing easy assessment across many task types. (2) **Agentic scaffolds:** Unlike existing benchmarks such as MLGym (17), `TimeSeriesGym` is agnostic to agent implementations, enabling a wide range of agent scaffolds to be integrated with minimal effort. (4) **Data sources:** `TimeSeriesGym` accommodates both Kaggle-style and non-Kaggle datasets (such as datasets in `TimeSeriesGym` Original challenges), making it straightforward for practitioners to introduce new datasets regardless of source or format[5]. Additionally, we offer clear and detailed documentation for adding new challenges to the benchmark, which has already enabled members of our community (outside our research team) to contribute a new challenge within 2 hours. Together, these design choices ensure that `TimeSeriesGym` can continuously grow and adapt as time series machine learning techniques continue to advance.

### 3.1 DESIGN CHOICES

**Focus on time series tasks.** We focused on time series modeling tasks for two key reasons. First, time series data is ubiquitous and critical in domains such as healthcare and economics, yet existing agentic AI benchmarks include relatively few time series challenges (Tab. 1). Second, compared to text and images, time series data require modest resources for storage and modeling, making `TimeSeriesGym` efficient to run. Moreover, modeling time-series data remains relatively underexplored outside specialized research communities, meaning that LLMs are less likely to have encountered such data and tasks during training. This characteristic, combined with the fact that `TimeSeriesGym` evaluates general machine learning skills, makes it an excellent testbed to evaluate AI agent capabilities. Although focused on time series, our benchmark can be be readily extended to other modalities, and already includes multimodal challenges, such as `TimeSeriesExam` (time series + text) and `OSIC Pulmonary Fibrosis Progression` (time series + images).

**How much freedom should the agents be given?** When designing challenges for `TimeSeriesGym`, we had to strike a fine balance between giving agents freedom to solve problems

---

[5]`TimeSeriesGym` is designed to be extensible while maintaining high standards of correctness, difficulty, and non-triviality. To support benchmark growth without compromising quality, we implement a category-specific quality-assurance pipeline, outlined in Sec H

creatively while keeping enough structure in place to allow for a precise and fine-grained evaluation. For example, in the `PTB-XL ECG Classification with Hyper-parameter Optimization` challenge, we *required* agents to use a PyTorch-based neural network and save their models, files and code before and after tuning. This allowed us to inspect models and code to check if the hyper-parameters changed, and measure how these changes improved performance.

**Agent-agnostic design.** `TimeSeriesGym` is agnostic to specific agent implementations. Following `MLE-bench` (3), it is easy to add new challenges and agentic scaffolds. To illustrate this flexibility, we include the *latest* implementations of 3 different scaffolds, `AIDE` (11), `MLAgentBench` (`MLAB`) (9), and `OpenHands` (21) with fundamentally different designs. Unlike `MLGym` (17), we do not advocate for a default agentic scaffold, as we believe that agent designs will continue to evolve and no single scaffold will work best for all ML engineering tasks.

# 4 EXPERIMENTS AND RESULTS

Table 2: **Main Results**. Each experiment was run with 3 random seeds, with results showing mean $\pm$ standard deviation. The table compares scaffold types (`AIDE`, `OpenHands`, `MLAB`), model choices (`GPT-4.1`, `o3`, `Claude 3.7`), resource allocations (4/50 to 12/150 hours/steps), and time utilization approaches. Key findings include: (1) `AIDE` achieves the best performance as a scaffold, (2) the reasoning model `o3` achieves significantly higher valid submission rates (94.4%) than other models, (3) `Claude 3.7` produces the most reasonable submissions (38.9%), (4) doubling time resources does not consistently improve performance, and (5) interestingly, removing step-wise reminders sometimes improves reasonable submission rates.

| `Lite` | Model | Resources (hours / steps) | Valid Submission (%) | Reasonable Submission (%) |
|---|---|---|---|---|
| | `MLAB` | | | |
| ✓ | + `gpt-4.1-2025-04-14` | 4 / 50 | $44.4 \pm 9.6$ | $27.8 \pm 9.6$ |
| | `OpenHands` | | | |
| ✓ | + `gpt-4.1-2025-04-14` | 4 / 50 | $44.4 \pm 19.3$ | $11.1 \pm 9.6$ |
| | `AIDE` | | | |
| ✓ | + `gpt-4.1-2025-04-14` | | $66.7 \pm 16.7$ | $27.8 \pm 9.6$ |
| | + `o3-2025-04-16` | | $\mathbf{94.4 \pm 9.6}$ | $33.3 \pm 0.0$ |
| | + `claude-3-7-sonnet-20250219` | 4 / 50 | $50.0 \pm 16.7$ | $38.9 \pm 19.3$ |
| | + deepseek-reasoner | | $11.1 \pm 9.6$ | $11.1 \pm 9.6$ |
| | + deepseek-chat | | $16.7 \pm 0.0$ | $16.7 \pm 0.0$ |
| ✗ | + `gpt-4.1-2025-04-14` | 4 / 50 | $58.6 \pm 7.6$ | $12.1 \pm 0$ |
| *Effect of Scaling Resources* | | | | |
| | | 4 / 50 | $66.7 \pm 16.7$ | $27.8 \pm 9.6$ |
| ✓ | + `gpt-4.1-2025-04-14` | 8 / 100 | $72.2 \pm 9.6$ | $22.2 \pm 9.6$ |
| | | 12 / 150 | $61.1 \pm 9.6$ | $\mathbf{50.0 \pm 0.0}$ |
| *Effective Utilization of Time* | | | | |
| ✓ | Step-wise reminder | 4 / 50 | $66.7 \pm 16.7$ | $27.8 \pm 9.6$ |
| | No reminder | | $55.6 \pm 9.6$ | $33.3 \pm 0.0$ |

**Setting.** We run agents in an Ubuntu 20.04 Docker container with all necessary resources (datasets, code repositories, etc.) and basic Python packages useful for ML engineering. Agents can access the internet and install additional packages as needed. For each challenge, agents have a maximum of 4 hours and 50 steps (9; 18; 17) and use a machine with 128 vCPUs, 503 GB RAM, 1.8 TiB SSD, and a single NVIDIA A100-SXM4-80GB GPU[6]. Unless otherwise specified, we repeat each experiment with 3 different seeds (0, 1, and 2) to calculate mean and standard deviation.

---

[6]In practice, agents share this machine as we run multiple challenges in parallel. This represents a realistic setting similar to how ML engineers routinely share computing resources. We found no instances where this sharing might have disadvantaged any agent.

**Cost.** On average, it cost us USD 63.00 to run `AIDE` with `gpt-4.1-2025-04-14` for a maximum of 4 hours and 50 steps on `TimeSeriesGym`. In contrast, the `Lite` benchmark was much more affordable at USD 8.00 per run. Given that `TimeSeriesGym -Lite` preserves coverage across domains and problem types while being much more time- and cost-effective, we conducted most of our experiments on `TimeSeriesGym -Lite` to accommodate our resource constraints.

**Metrics.** We report the raw scores achieved by `AIDE` on every challenge (Tab. 13). Although these scores are useful for tracking progress on individual challenges, they cannot be easily combined across different challenges. To measure the performance of agents at a high level, we report two key metrics: the percentage of challenges where the agent made a (1) *valid*, and (2) *reasonable* submission (Tab. 2). A submission is valid if the grader returns any non-null score. What counts as a reasonable attempt varies by challenge type. For Kaggle challenges, we define it as scoring above median on the competition's public leaderboard. For the remaining challenges, we consider a submission reasonable if it made a genuine[7] modeling attempt rather than hallucinating an output that matches the submission format. For example, simply loading and re-saving the provided sample submission file without any model inference or data processing is deemed unreasonable.

## 4.1 OBSERVATIONS

**`AIDE` is the better open-source scaffold.** We evaluated `GPT-4.1` (`gpt-4.1-2025-04-14`) with three open-source scaffolds: `AIDE` (11), `MLAB` (9) and `OpenHands` (21). Following `MLE-bench`, we make minor modifications to adapt these scaffolds to our benchmark (see Appendix C). Results in Tab. 2 confirm prior findings: `AIDE` with `GPT-4.1` yields the highest proportion of valid (66.7%) and reasonable (27.8%) submissions. This is expected as `AIDE` is specifically designed for data science tasks, which account for the majority of `TimeSeriesGym` challenges.

**Reasoning models produce substantially more valid submissions.** To identify the best base model, we evaluated the strongest scaffold (`AIDE`) with two state-of-the-art proprietary LLMs: `GPT-4.1` and `Claude 3.7` (`claude-3-7-sonnet-20250219`), and a reasoning model `o3` (`o3-2025-04-16`). As shown in Tab. 2, our experiments on `TimeSeriesGym-Lite` revealed that `o3` created significantly more valid submissions than other models, while `Claude 3.7` produced the highest number of reasonable attempts (38.9%). We noticed a significant gap between valid and reasonable submissions for `o3`. While `o3` can generate valid submission files for most challenges by following the instructions provided, it was also prone to hallucination. In some cases, it produced a submission file without any genuine modeling attempt (e.g., simply outputting a zero array). An illustration of this failure mode is provided in Appendix F.4.

**Challenges in `TimeSeriesGym` are hard for state-of-the-art agents.** We tested `AIDE` with `GPT-4.1` on all `TimeSeriesGym` challenges (see Tab. 13) and found poor overall performance. The agent produced valid submissions for only 58.6% of challenges and reasonable submissions for only 12.1% on average. We found that the agent especially struggled with `TimeSeriesGym` original challenges, where it only produced valid submissions for 5 out of 14 challenges. These results show that even the best agents struggle with ML engineering tasks, particularly those that go beyond standard Kaggle data science challenges and involve working with multi-file code repositories.

**Agents do not improve with more time.** We wondered if the agents perform poorly on `TimeSeriesGym` simply because they need more time. To test this, we ran `AIDE` with `GPT-4.1` on `TimeSeriesGym -Lite` and gave it 2–3× more hours and steps per challenge. Our results show that extra time does not always improve performance (Tab. 14). Even with the maximum time (12 hours and 150 steps), the agent only made reasonable submissions in about 50% of challenges–not very impressive given that the bar for a "reasonable" submission is quite low.

**Agents do not utilize time effectively.** We suspected that agents do not improve with more time because they do not use it well. To test this idea, we compared two settings: the default approach of reminding the agent about remaining time (and steps) before each step, versus removing these reminders completely. Surprisingly, we did not find significant differences between these settings. In fact, agents without time reminders produced more reasonable submissions. This may suggest that

---

[7]We assess this by manually inspecting whether a modeling attempt was made. Since this is inherently a subjective judgment, reasonable attempts are reported ONLY as a study of agentic skills. The benchmark score itself ultimately depends on the objective metric we defined for each challenge.

agents do not use their time wisely– they tend to rush toward solutions instead of carefully exploring promising options, especially towards the end of the experiment. This raises important research questions about how to design agents that use their time and resources more strategically.

**Frontier LLM challenges.** Since frontier LLMs are pretrained on large-scale public data, there is a risk that they may have encountered and memorized content from public challenges, e.g., online Kaggle competition discussions or solutions, which can potentially inflate benchmark performance and limit its generalizability. To assess this risk, we followed the approach used by `MLE-bench` to measure `GPT-4.1`'s familiarity with `TimeSeriesGym` challenges and compared the familiarity score distribution to that of `MLE-bench` (see Appendix E.4). `GPT-4.1` exhibited a similar level of familiarity with `TimeSeriesGym` challenges as with `MLE-bench` challenges (with Kolmogorov–Smirnov (KS) Test (15) p-value = 0.363, indicating no significant difference). Given that `MLE-bench` found no systematic impact of LLM familiarity on experiment results, `GPT-4.1`'s familiarity with `TimeSeriesGym` is within a reasonable range and does not compromise its integrity.

**`TimeSeriesGym` can be an effective diagnostic tool for agentic skill development.** We stratify the results in Tab. 2 in terms of core ML skills that each challenge tests (Tab. 3). For example, we found that agents struggle in hyper-parameter tuning– 2 out of the 3 scaffolds (`OpenHands` and `MLAB`) failed to produce valid submissions. While all scaffolds perform similarly on code migration, `AIDE` achieved the best performance on handling missing data, likely due to its specific design for data science tasks. A similar analysis for base models is provided in Appendix E.3.

Table 3: Performance of agent scaffolds with `GPT-4.1` (Best@3 / Avg@3) on different ML skills. Arrows indicate whether lower ($\downarrow$) or higher ($\uparrow$) values are better. Agents struggle with hyper-parameter tuning.

| ML Skill | Metric | AIDE | OpenHands | MLAB |
|---|---|---|---|---|
| Handling Data Missingness | Root Mean Square Percentage Error ($\downarrow$) | 0.33/0.33 | 0.64/0.64 | 0.42/0.89 |
| Code Migration | Percentage of Test Cases Passed ($\uparrow$) | 0.56/0.56 | 0.56/0.44 | 0.56/0.56 |
| Hyper-parameter Tuning | Improvement in Accuracy ($\uparrow$) | 0.08/0.03 | N/A / N/A | N/A / N/A |

**Summary.** This section provides a focused illustration of how `TimeSeriesGym` enables efficient and cost-effective experimentation with AI agents, helping researchers uncover actionable insights about agent capabilities and limitations. Our findings demonstrate `TimeSeriesGym`'s value for advancing generic ML engineering agents.

## 5 DISCUSSION, OPEN QUESTIONS AND OPPORTUNITIES

**Key limitations of existing scaffolds.** Agentic scaffolds such as `AIDE` and `OpenHands` provide structured workflows that **excel in single-shot, self-contained benchmarks** (e.g., Kaggle competitions) but reveal **significant limitations in repository-level challenges** that require multiple file edits and iterative reasoning. `AIDE`'s *one-step solution strategy* and *fixed action set*—restricted to predefined operations such as "data preview" when debugging—often lead to unsuccessful attempts in large codebases, as the agent's attention is diluted across irrelevant files and fails to identify critical information. Conversely, `OpenHands` supports multi-step trajectories yet suffers from a **greedy exploitation bias**: it commits fully to a single approach without exploring alternative solution paths or revisiting earlier decisions when trajectories prove unfruitful. The planning algorithm of the `CodeAct` agent used by `OpenHands` is similarly *greedy and short-horizon*, limiting adaptation to complex multistage development workflows. These findings highlight the need for **more adaptive scaffolds** that dynamically expand their action repertoire, balance exploration and exploitation through parallel solution threads, and support nested workflows reflective of real-world machine learning engineering tasks. We provide illustrations of agent failures in Appendix F.

**Data leakage and plagiarism.** In designing `TimeSeriesGym`, we identify two key risks related to data leakage and plagiarism that could compromise the integrity of the benchmark: (1) **Pretraining contamination:** Current LLMs may have been exposed to public content from existing challenges (e.g., Kaggle competitions), including task descriptions, data, or shared solutions. This can lead to memorization and inflated performance that overstates agents' true capabilities, and (2) **Future LLM**

**leakage:** Once the benchmark is public, future LLMs may be pretrained on its content, making the benchmark less effective in evaluating real generalization.

To address such risks, we present both empirical findings and mitigation strategies. For case (1), we have two key observations. First, in both Kaggle-based and original challenges in `TimeSeriesGym`, agents either performed poorly or did not produce valid output, suggesting minimal benefit from any potential LLM contamination. Second, we conducted a formal analysis using available tools to assess agents' familiarity with all competitions in this benchmark. The results show no evidence of systematic prior exposure or memorization, further supporting the integrity of the benchmark in its current state. For case (2), the scalability of `TimeSeriesGym` enables efficient generation of new challenges and skill-specific variations. This allows the benchmark to evolve continuously and remain effective even if the current version is eventually included in future LLM pretraining.

Finally, we raise a broader question around plagiarism and code reuse. Several `TimeSeriesGym` challenges, such as leveraging `MOMENT` (6) for anomaly detection, require agents to use existing code repositories to solve open-ended ML tasks, making plagiarism difficult to assess. For example, if an agent cites the code it uses, should it be considered plagiarism or appropriate reuse, similar to how human ML practitioners build on public code with proper reference? As the ability to effectively and properly leverage existing resources is important in real-world ML practice, we believe that it is crucial to develop clear, legally correct definitions and evaluation criteria for data contamination and plagiarism in the context of AI agents. We highlight this as an important direction for future work.

**Defining and measuring success.** What does it mean for an agent to be successful? For Kaggle tasks, while comparing an agent's performance against human leaderboards seems intuitive, it presents challenges. `TimeSeriesGym` utilizes different training and testing splits and re-implements the grading mechanisms from the original Kaggle competitions (as original Kaggle test sets are private), making direct leaderboard comparisons potentially misleading. Additionally, in challenges like code migration, real-world utility does not always require perfect, bug-free code: partial, buggy solutions may still accelerate development when iterated by human engineers. Thus, our current evaluation approaches have inherent limitations.

We propose several desiderata for improving the success metrics. These metrics should be rigorous and objective, yet flexible enough to preserve agent creativity and autonomy. They should also yield actionable insights, helping identify specific deficiencies and guide future improvements in agent design. In this work, we take a step in this direction by enabling skill-based and holistic evaluations, offering a more comprehensive understanding of agent capabilities and limitations. Moving forward, we believe that the development of robust, holistic and diagnostic success metrics remains an important research direction and requires community discussion.

**Optimal resource allocation.** Consistent with previous work, agents were given 4 hours and 50 steps to solve each challenge - but is this sufficient? Alternative frameworks like `MLE-bench` provide substantially more resources (24 hours and approximately 2000 steps). Our scaling experiments, which gave agents up to 12 hours and 150 steps for a subset of challenges, did not reveal significant performance improvements. Therefore, we believe that further increasing resources is an option, but practical academic budget constraints make such approaches largely infeasible. This raises important questions about how to balance resource limitations with fair opportunities to assess AI agents.

**Societal impact.** AI agents promise to substantially reduce manual effort in ML engineering while expanding the productivity and accessibility of ML tools. This automation presents several social implications worth considering. First, by lowering technical barriers, these agents could democratize ML capabilities, allowing users without an extensive programming background to leverage advanced analytics. Second, automated ML workflows can accelerate scientific discovery across multiple domains, including healthcare, climate science, and materials research. However, several challenges require careful attention from the community. The primary concern is proper attribution when agents repurpose existing code, potentially obscuring original authorship and violating licenses. Furthermore, automated ML systems can perpetuate or amplify existing biases in training data without human oversight. Furthermore, these agents might generate plausible but flawed solutions that appear correct to non-experts, leading to undetected errors in critical applications. The resource-intensive nature of running sophisticated agents could also exacerbate computational divides between well-resourced and under-resourced organizations. As we advance agent capabilities through benchmarks

like `TimeSeriesGym`, the community must simultaneously develop frameworks for responsible deployment that address these challenges while maximizing societal benefits.

## 6 CONCLUSION

We propose `TimeSeriesGym`, a scalable and agent-agnostic benchmarking framework for evaluating AI agents (with various scaffoldings) on time series ML engineering tasks. By curating tasks that reflect real-world ML practice from diverse sources, enabling scalable task generation, and supporting multimodal, skill-based, holistic evaluation, `TimeSeriesGym` provides a practical and extensible testbed for advancing AI agents in ML engineering. Our experiments show that while frontier LLMs combined with `AIDE` scaffolding (11) can achieve moderate to high success rates in producing valid submissions, they still do not generate reasonable solutions, particularly on *TimeSeriesGym original* challenges that emulate the complexity of real-world time series tasks. This highlights current limitations in agent capabilities to effectively understand and solve realistic time series tasks. By open-sourcing `TimeSeriesGym`, we aim to facilitate a deeper understanding of the ML engineering capabilities of AI agents, provide actionable insights on future development, and support the collection of agent interaction trajectories to drive continuous improvement of AI agents through post-training and refinement.

## REPRODUCIBILITY STATEMENT

We provide `TimeSeriesGym` as an open-source project under the permissive MIT License: `https://anonymous.4open.science/r/TimeSeriesGym-9CF6/`. The repository includes detailed documentation on running experiments, adding new challenges, and incorporating different agentic scaffolds. Tab. 4 lists all challenges in `TimeSeriesGym`, while Tab. 5 provides their sources and licenses. We describe our exact experimental settings and compute resources in Sec. 4, with scaffold hyperparameters detailed in Tab. 11. The cost to run each experiment is reported in Tab. 7.

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

## A  TIMESERIESGYM CHALLENGES

| Challenge | Problem | Domain | Skills | Evaluation Metric |
|---|---|---|---|---|
| **Kaggle Challenges** | | | | |
| AMP-Parkinson's Disease Progression Prediction | Time-to-Event Regression | Healthcare | | Symmetric Mean Absolute Percentage Error |
| ASHRAE - Great Energy Predictor III | Forecasting | Energy | | Root Mean Square Logarithmic Error |
| Child Mind Institute– Detect Sleep States | Classification | Healthcare | | Event Detection Average Precision |
| Google Brain - Ventilator Pressure Prediction | Regression | Healthcare | Data Handling (Dealing with Missing Values, Utilize Multi-Source Data) | Mean Absolute Error |
| G2Net Gravitational Wave Detection | Classification | Geology | | Area Under ROC Curve |
| HMS - Harmful Brain Activity Classification | Classification | Healthcare | | KL Divergence |
| LANL Earthquake Prediction | Time-to-Event Regression | Geology | Modeling (Hyper-parameter Tuning & Model Selection) | Mean Absolute Error |
| M5 Forecasting - Accuracy | Forecasting | Sales | | Weighted Root Mean Squared Scaled Error |
| Online Product Sales | Forecasting | Sales | | Root Mean Square Logarithmic Error |
| Optiver Realized Volatility Prediction | Forecasting | Finance | | Root Mean Square Percentage Error |
| OSIC Pulmonary Fibrosis Progression | Forecasting | Healthcare | | Laplace Log Likelihood |
| Recruit Restaurant Visitor Forecasting | Forecasting | Sales | | Root Mean Square Logarithmic Error |
| Sberbank Russian Housing Market | Forecasting | Housing | | Root Mean Square Logarithmic Error |
| **TimeSeriesGym Originals** | | | | |
| Convert ResNet TensorFlow Implementation to PyTorch | Classification | | | |
| Convert STOMP Algorithm Implementation in R to Python | Data Mining | Algorithm | Code Migration | Custom Code Grading |
| Evaluate MOIRAI time series foundation model on the Context Is Key (CiK) benchmark | Context-aided Forecasting | Climatology, Economics, Energy, Mechanics, Public Safety, Retail, Synthetic, Transportation | | Resolved (Binary) |
| Evaluate Chronos time series foundation model on the NN5 dataset within Context Is Key (CiK) benchmark | | | | |
| Implement & Evaluate CSDI to Impute PM2.5 Data | Imputation | Weather | | Mean Absolute Error |
| Train & Evaluate CSDI to Impute PM2.5 Data | | | | |
| GIFT-EVAL: A Benchmark for General Time Series Forecasting Model Evaluation | Forecasting | Nature, Web, CloudOps, Economics/Finance, Energy, Sales, Transportation, Healthcare, Gait, Energy, Synthetic, Devices | Modeling (Using Research Code) | Mean Absolute Percentage Error |
| Hexagon ML UCR Time Series Anomaly Detection | Anomaly Detection | | | Adjusted Best F1 Score |
| Long Horizon Time Series Forecasting using Time Series Library | Forecasting | Energy, Epidemiology, Finance, Transportation, Weather | | Mean Squarred Error |
| Long-Horizon Weather Forecasting using Time Series Library's Itransformer | Forecasting | Weather | | Exact Match |
| MIT-BIH ECG Arrhythmia Detection | Classification | Healthcare | | Accuracy |
| MOMENT for Anomaly Detection on UCR datasets | Anomaly Detection | Healthcare, Gait, Energy, Synthetic, Devices | | Exact Match |
| PTB-XL ECG Classification | Classification | Healthcare | | Accuracy |
| TimeSeriesExam: A Time Series Understanding Exam | Time Series Understanding | Synthetic | Time Series Understanding | Accuracy |
| **Derived Challenges** | | | | |
| Google Brain - Ventilator Pressure Prediction | Regression | Healthcare | Data Handling (Dealing with missing data) | Mean Absolute Error |
| Improve PTB-XL ECG Classification Code | Classification | Healthcare | Code Enhancement (Experiment Tracking, Readability, Reproducibility) | |
| MIT-BIH Arrhythmia Detection with Weak Supervision | Classification | Healthcare | Data Handling (Labeling) | Accuracy |
| Optiver Realized Volatility Prediction | Forecasting | Finance | Data Handling (Dealing with missing data) | Root Mean Square Percentage Error |
| Optiver Realized Volatility Prediction with Hyper-parameter Optimization | Forecasting | Finance | | Improvement in Root Mean Square Percentage Error |
| PTB-XL ECG Classification with Hyperparameter Optimization | Classification | Healthcare | Modeling (Hyper-parameter Tuning & Model Selection) | Improvement in Accuracy |

Table 4: This table presents the `TimeSeriesGym` benchmark's diverse collection of time series challenges across three categories: Kaggle Challenges, `TimeSeriesGym` Originals, and Derived Challenges. The challenges span multiple domains (healthcare, finance, energy, weather, transportation), problem types (classification, regression, forecasting, anomaly detection), and required skills (data handling, model building, code migration). Each challenge uses appropriate evaluation metrics for its task type. The benchmark combines established Kaggle competitions with novel custom tasks, creating a comprehensive testbed for evaluating ML engineering agents across realistic scenarios that practitioners face in real-world applications.

| Challenge | Source | License |
|---|---|---|
| ***Kaggle Challenges*** | | |
| AMP-Parkinson's Disease Progression Prediction | Kaggle | Subject to Competition Rules |
| ASHRAE - Great Energy Predictor III | Kaggle | Subject to Competition Rules |
| Child Mind Institute– Detect Sleep States | Kaggle | CC BY-NC-SA 4.0 |
| Google Brain - Ventilator Pressure Prediction | Kaggle | Subject to Competition Rules |
| G2Net Gravitational Wave Detection | Kaggle | Subject to Competition Rules |
| HMS - Harmful Brain Activity Classification | Kaggle | CC BY-NC 4.0 |
| LANL Earthquake Prediction | Kaggle | Subject to Competition Rules |
| M5 Forecasting - Accuracy | Kaggle | Subject to Competition Rules |
| Online Product Sales | Kaggle | Subject to Competition Rules |
| Optiver Realized Volatility Prediction | Kaggle | Subject to Competition Rules |
| OSIC Pulmonary Fibrosis Progression | Kaggle | Subject to Competition Rules |
| Recruit Restaurant Visitor Forecasting | Kaggle | Subject to Competition Rules |
| Sberbank Russian Housing Market | Kaggle | Subject to Competition Rules |
| ***TimeSeriesGym Originals*** | | |
| Convert ResNet TensorFlow Implementation to PyTorch | GitHub | GNU General Public License v3.0 |
| Convert STOMP Algorithm Implementation in R to Python | GitHub | Apache License 2.0 |
| Evaluate MOIRAI time series foundation model on the Context Is Key (CiK) benchmark | GitHub | Apache License 2.0 |
| Evaluate Chronos time series foundation model on the NN5 dataset within Context Is Key (CiK) benchmark | – | Apache License 2.0 |
| Implement & Evaluate CSDI to Impute PM2.5 Data | GitHub | MIT License |
| Train & Evaluate CSDI to Impute PM2.5 Data | – | MIT License |
| GIFT-EVAL: A Benchmark for General Time Series Forecasting Model Evaluation | GitHub | Apache License 2.0 |
| Hexagon ML UCR Time Series Anomaly Detection | UCR | Not available |
| Long Horizon Time Series Forecasting using Time Series Library | GitHub | MIT License |
| Long-Horizon Weather Forecasting using Time Series Library's Itransformer | – | MIT License |
| MIT-BIH ECG Arrhythmia Detection | PhysioNet | Open Data Commons Attribution License v1.0 |
| MOMENT for Anomaly Detection on UCR datasets | GitHub | MIT License |
| PTB-XL ECG Classification | PhysioNet | Creative Commons Attribution 4.0 International Public License |
| TimeSeriesExam: A Time Series Understanding Exam | Hugging Face | MIT License |

Table 5: This table provides transparency about the source and licensing information for each challenge in the `TimeSeriesGym` benchmark. For the Kaggle challenges, most are subject to Kaggle's competition rules, with a few under Creative Commons licenses. The `TimeSeriesGym` Original challenges come from diverse sources including GitHub repositories, HuggingFace, etc. with various open-source licenses (Apache, MIT, GPL, Creative Commons). This diversity of sources and licenses demonstrates the benchmark's foundation in accessible, reusable datasets and code while ensuring proper attribution and compliance with original creators' terms.

Table 6: `TimeSeriesGym`–Lite is a streamlined collection of six diverse time series challenges carefully selected to evaluate AI agents while balancing comprehensiveness with efficiency. The challenges cover essential ML engineering skills including basic data science, handling missing/multi-source data, code migration, hyperparameter optimization, modeling using research code, and data labeling. The collection spans multiple domains (healthcare, finance, algorithms) and various time series tasks (classification, forecasting, anomaly detection, code migration). This cost-effective subset allows researchers to quickly benchmark agent capabilities across critical ML engineering skills without the resource requirements of the full `TimeSeriesGym` benchmark.

| Challenge | Required Skills | Time Series Task | Domain |
|---|---|---|---|
| Child Mind Institute - Detect Sleep States | Basic data science (data handling and modeling) | Classification | Healthcare |
| Optiver Realized Volatility Prediction | Handling missing and multi-source data | Forecasting | Finance |
| Convert ResNet TensorFlow implementation to PyTorch | Classification | Code Migration | Algorithm |
| PTB-XL ECG Classification | Hyperparameter optimization & model selection | Classification | Healthcare |
| MOMENT Anomaly Score Calculation | Modeling (Using research code) | Anomaly Detection | Healthcare, Gait, Synthetic, Energy, Devices |
| MIT-BIH Arrhythmia Detection | Data labeling | Classification | Healthcare |

Table 7: Average cost to run experiments on a single seed in the default evaluation setup *i.e.* AIDE with `gpt-4.1-2025-04-14` with a maximum of 4 hours and 50 steps.

| Benchmark | Averge Cost (USD) |
|---|---|
| TimeSeriesGym | 62.12 |
| TimeSeriesGym –Lite | 7.96 |

Table 8: The 100 shortlisted Kaggle competitions. Competitions marked with * denote that the data is no longer available.

| Competition | # Participants | Reward | Category |
|---|---|---|---|
| M5 Forecasting - Accuracy | 7022 | 50,000 | Featured |
| LANL Earthquake Prediction | 5454 | 50,000 | Research |
| Jane Street Market Prediction* | 4884 | 100,000 | Featured |
| Optiver Realized Volatility Prediction | 4395 | 100,000 | Featured |
| Optiver - Trading at the Close | 4374 | 100,000 | Featured |
| ASHRAE - Great Energy Predictor III | 4342 | 25,000 | Featured |
| Zillow Prize: Zillow's Home Value Prediction (Zestimate) | 4241 | 1,200,000 | Featured |
| GoDaddy - Microbusiness Density Forecasting | 3834 | 60,000 | Featured |
| Rossmann Store Sales | 3735 | 35,000 | Featured |
| Sberbank Russian Housing Market | 3658 | 25,000 | Featured |
| HMS - Harmful Brain Activity Classification | 3507 | 50,000 | Research |
| Google Brain - Ventilator Pressure Prediction | 3118 | 7,500 | Research |
| University of Liverpool - Ion Switching | 3004 | 25,000 | Research |
| Ubiquant Market Prediction* | 2949 | 100,000 | Featured |
| Enefit - Predict Energy Behavior of Prosumers | 2715 | 50,000 | Featured |
| OSIC Pulmonary Fibrosis Progression | 2530 | 55,000 | Featured |
| Child Mind Institute - Detect Sleep States | 2436 | 50,000 | Featured |
| Recruit Restaurant Visitor Forecasting | 2426 | 25,000 | Featured |

Continued on next page

| Competition | # Participants | Reward | Category |
|---|---|---|---|
| G-Research Crypto Forecasting | 2398 | 125,000 | Featured |
| Two Sigma Financial Modeling Challenge* | 2317 | 100,000 | Featured |
| Grupo Bimbo Inventory Demand | 2263 | 25,000 | Featured |
| AMP®-Parkinson's Disease Progression Prediction | 2197 | 60,000 | Featured |
| Corporación Favorita Grocery Sales Forecasting | 1868 | 30,000 | Featured |
| Driver Telematics Analysis | 1861 | 30,000 | Featured |
| Parkinson's Freezing of Gait Prediction | 1688 | 100,000 | Research |
| COVID19 Global Forecasting (Week 1) | 640 | 0 | Research |
| Heritage Health Prize* | 1656 | 500,000 | Featured |
| Cornell Birdcall Identification | 1630 | 25,000 | Research |
| TensorFlow Speech Recognition Challenge | 1591 | 25,000 | Featured |
| Google - American Sign Language Fingerspelling Recognition | 1529 | 200,000 | Research |
| G2Net Gravitational Wave Detection | 1501 | 15,000 | Research |
| COVID19 Global Forecasting (Week 4) | 388 | 0 | Research |
| Indoor Location & Navigation | 1446 | 10,000 | Research |
| West Nile Virus Prediction | 1445 | 40,000 | Featured |
| BirdCLEF 2023 | 1397 | 50,000 | Research |
| Rainforest Connection Species Audio Detection | 1385 | 15,000 | Research |
| COVID19 Global Forecasting (Week 3) | 290 | 0 | Research |
| COVID19 Global Forecasting (Week 2) | 263 | 0 | Research |
| Google Analytics Customer Revenue Prediction | 1369 | 45,000 | Featured |
| COVID19 Local US-CA Forecasting (Week 1) | 216 | 0 | Research |
| Google - Isolated Sign Language Recognition | 1340 | 100,000 | Research |
| 1st and Future - Player Contact Detection | 1334 | 100,000 | Featured |
| JPX Tokyo Stock Exchange Prediction | 1324 | 63,000 | Featured |
| Google Research Football with Manchester City F.C. | 1288 | 6,000 | Featured |
| Lyft Motion Prediction for Autonomous Vehicles | 1254 | 30,000 | Featured |
| BirdCLEF 2024 | 1198 | 50,000 | Research |
| COVID19 Global Forecasting (Week 5) | 93 | 0 | Research |
| G2Net Detecting Continuous Gravitational Waves | 1181 | 25,000 | Research |
| Peking University/Baidu - Autonomous Driving | 1105 | 25,000 | Featured |
| iWildcam 2021 - FGVC8 | 65 | 0 | Research |
| Eye Movements Verification and Identification Competition | 50 | 0 | Research |
| M5 Forecasting - Uncertainty | 1101 | 50,000 | Featured |
| March Machine Learning Mania 2023 | 1098 | 50,000 | Featured |
| Multi-label Bird Species Classification - NIPS 2013 | 39 | 0 | Research |
| Google Cloud & NCAA® ML Competition 2018-Men's | 1061 | 50,000 | Featured |
| iWildCam 2022 - FGVC9 | 29 | 0 | Research |
| NFL Health & Safety - Helmet Assignment | 1028 | 100,000 | Featured |
| March Machine Learning Mania 2022 - Men's | 1025 | 25,000 | Featured |
| BirdCLEF 2022 | 1009 | 10,000 | Research |
| BirdCLEF 2021 - Birdcall Identification | 1001 | 5,000 | Research |
| Google Smartphone Decimeter Challenge | 985 | 10,000 | Research |
| SETI Breakthrough Listen - E.T. Signal Search | 979 | 15,000 | Research |
| LEAP - Atmospheric Physics using AI (ClimSim) | 877 | 50,000 | Research |
| Bengali.AI Speech Recognition | 866 | 53,000 | Research |
| The Winton Stock Market Challenge | 829 | 50,000 | Featured |
| Two Sigma: Using News to Predict Stock Movements | 813 | 100,000 | Featured |
| Accelerometer Biometric Competition | 770 | 5,000 | Research |
| How Much Did It Rain? II | 691 | 500 | Research |
| Google Smartphone Decimeter Challenge 2022 | 684 | 10,000 | Research |
| American Epilepsy Society Seizure Prediction Challenge | 653 | 25,000 | Research |
| Melbourne University AES/MathWorks/NIH Seizure Prediction | 645 | 20,000 | Research |
| DFL - Bundesliga Data Shootout | 610 | 25,000 | Featured |
| NFL 1st and Future - Impact Detection | 573 | 75,000 | Featured |

| Competition | # Participants | Reward | Category |
|---|---|---|---|
| Kore 2022 | 537 | 15,000 | Featured |
| MLB Player Digital Engagement Forecasting | 495 | 50,000 | Featured |
| ECML/PKDD 15: Taxi Trajectory Prediction (I) | 459 | 250 | Research |
| Grasp-and-Lift EEG Detection | 451 | 10,000 | Research |
| The Big Data Combine Engineered by BattleFin | 424 | 18,500 | Research |
| Web Traffic Time Series Forecasting | 424 | 25,000 | Research |
| Draper Satellite Image Chronology | 422 | 75,000 | Featured |
| ECML/PKDD 15: Taxi Trip Time Prediction (II) | 418 | 250 | Research |
| Online Product Sales | 412 | 22,500 | Featured |
| Halite by Two Sigma | 1291 | 0 | Featured |
| RTA Freeway Travel Time Prediction | 376 | 10,000 | Featured |
| How Much Did It Rain? | 349 | 500 | Research |
| The 3rd YouTube-8M Video Understanding Challenge | 340 | 25,000 | Research |
| Benchmark Bond Trade Price Challenge | 316 | 17,500 | Featured |
| BCI Challenge @ NER 2015 | 311 | 1,000 | Research |
| The Marinexplore and Cornell University Whale Detection Challenge | 309 | 10,000 | Featured |
| DecMeg2014 - Decoding the Human Brain | 301 | 5,000 | Research |
| U.S. Census Return Rate Challenge | 290 | 25,000 | Featured |
| dunnhumby's Shopper Challenge | 287 | 10,000 | Featured |
| KDD Cup 2012, Track 2 | 275 | 8,000 | Featured |
| UPenn and Mayo Clinic's Seizure Detection Challenge | 241 | 8,000 | Research |
| GE Flight Quest | 234 | 250,000 | Featured |
| The Random Number Grand Challenge | 206 | 1,000 | Featured |
| AMS 2013-2014 Solar Energy Prediction Contest | 199 | 1,000 | Research |
| Global Energy Forecasting Competition 2012 - Wind Forecasting | 197 | 7,500 | Research |
| Belkin Energy Disaggregation Competition | 194 | 25,000 | Featured |
| CVPR 2018 WAD Video Segmentation Challenge | 188 | 2,500 | Research |

# B    REPRESENTATIVENESS OF TIMESERIESGYM-LITE

To ensure that `TimeSeriesGym-Lite` provides broad coverage of machine learning capabilities while enabling low-cost evaluation, we construct the subset by selecting a diverse set of tasks from the full `TimeSeriesGym` benchmark. We further validate this selection through statistical comparisons of domain coverage and task difficulty distributions between the two benchmarks.

**Domain Coverage.**    Table 9 summarizes the domain distributions of `TimeSeriesGym` and `TimeSeriesGym-Lite`. A Chi-square test indicates no significant difference between the two distributions ($p = 0.92$, $\chi^2 = 1.99$), demonstrating that the Lite subset maintains the same domain diversity as the full benchmark.

| Domain | TimeSeriesGym | TimeSeriesGym-Lite |
|---|---|---|
| Healthcare | 11 | 3 |
| Multi-domain | 7 | 1 |
| Commerce & Finance | 6 | 1 |
| Weather | 3 | 0 |
| Geology | 2 | 0 |
| Housing | 1 | 0 |
| Energy | 1 | 0 |

Table 9: Domain distributions of `TimeSeriesGym` and `TimeSeriesGym-Lite`.

**Task Difficulty.**    We additionally compare task difficulty levels between the two benchmarks, shown in Table 10. A Chi-square test again reveals no significant difference ($p = 0.95$, $\chi^2 = 0.09$), indicating that the Lite subset preserves the difficulty profile of the full benchmark.

| Difficulty | TimeSeriesGym | TimeSeriesGym-Lite |
|---|---|---|
| Low | 13 | 2 |
| Medium | 15 | 3 |
| High | 5 | 1 |

Table 10: Difficulty distributions of `TimeSeriesGym` and `TimeSeriesGym-Lite`.

**Summary.**    These statistical tests confirm that `TimeSeriesGym-Lite` is a representative subset of `TimeSeriesGym` in both domain diversity and task difficulty. This ensures that the Lite benchmark provides reliable, low-cost evaluation while preserving the characteristics of the full task suite.

# C IMPLEMENTATION DETAILS FOR SCAFFOLDS

Table 11: Scaffold hyperparameters. `$TARGET_MODEL` denotes the model being evaluated.

| AIDE | |
| --- | --- |
| `agent.code.model` | `$TARGET_MODEL` |
| `agent.feedback.model` | `gpt-4.1-2025-04-14` |
| `agent.steps` | `50` |
| `agent.search.max_debug_depth` | `20` |
| `agent.search.debug_prob` | `1` |
| `agent.time_limit` | `14400` |
| `exec.timeout` | `32400` |
| **OpenHands** | |
| `agent` | `CodeActAgent` |
| `model` | `$TARGET_MODEL` |
| `max_time_in_hours` | `4` |
| `max_steps` | `50` |
| **MLAgentBench (MLAB)** | |
| `llm-name` | `$TARGET_MODEL` |
| `edit-script-llm-name` | `$TARGET_MODEL` |
| `fast-llm-name` | `gpt-4.0-mini` |
| `max-steps` | `50` |
| `max-time` | `14400` |

Tab. 11 summarizes the hyperparameters used for the three scaffolds: `AIDE` (11), `OpenHands`' `CodeAct` (21), and `MLAgentBench` (9). We did not directly use the MLE-bench (3) modifications to the agentic scaffolds for two main reasons. First, the official scaffold implementation has undergone updates since MLE-bench. Second, certain changes made in MLE-bench were not applicable to our benchmark, which involves more diverse modes of evaluation. Therefore, we modified the latest version of the agentic scaffolds to enhance robustness, improve execution stability, and support a broader range of competitions.

## C.1 AIDE

We forked the original `AIDE` repository in May 2025 and added useful changes from the MLE-bench project. These include better API calls and support for more API providers. Our key modifications are summarized below:

1. Updated instructions to cover all types of tasks, not just Kaggle challenges
2. Modified the prompt to work with different file types, since our tasks use many input/output formats
3. Fixed the interpreter handling to prevent timeouts and system hangs

## C.2 OPENHANDS

We forked the original `OpenHands` repository from tag `v0.34.0` (May 2025). We reduced the RAM allowance to 10 GiB (from 100 GiB) as we did not observe any memory-related issues during our tests.

## C.3 MLAGENTBENCH

We forked the MLE-bench version of `MLAgentBench` with no modifications.

Figure 2: The prompt we use to initialize all scaffolds, adapted from `MLE-bench` (3).

```
# Machine Learning Engineer Challenge: TimeSeriesGym Competition

## Your Role
You are a Machine Learning Engineer tasked with solving complex ML problems in the
↪  TimeSeriesGym offline competition. This competition tests your ability to understand
↪  problems, follow instructions, and build/train/evaluate ML models.

## Core Requirements
- You must solve each task by training a model and generating predictions on the test set
- Hand-labeling the data to produce submissions is strictly prohibited

## Environment Setup

### Critical Files and Locations
- **Competition Instructions**: Located at `/home/data/description.md` - This is your primary
↪  reference for understanding the specific problem
- **Dataset**: All data files are in `/home/data/` directory
- **Submission Path**: Your final submission MUST be saved to `/home/submission/` in the
↪  format specified in the instructions
- **Environment Instructions**: These general instructions can be found at
↪  `/home/instructions.txt`

### Test Dataset Information
- We've created our own train/test splits for evaluation purposes
- The data in `/home/data/` may differ slightly from what's described in the competition
↪  instructions
- Use the provided train/test splits for your work

## Submission Instructions

### File Format and Location
- Your submission MUST be saved to the exact path: `/home/submission/`
- The file format must strictly follow the specifications in `description.md`
- Failing to use the correct path or format will result in a failed evaluation

### Validation Process
1. Before finalizing your submission, validate its format using the validation server
2. Use the command: `curl -X POST -F "file=@${SUBMISSION_FILE}"
↪  http://localhost:5000/validate`
3. Alternatively, use the provided script: `/home/validate_submission.sh`
4. The validator will confirm if your submission format is valid (but won't provide a score)
5. Make corrections if needed and revalidate until you receive a confirmation

### Final Submission
- Once validated, ensure your final submission file remains at `/home/submission/`
- Only the last file in this location will be evaluated
- Double-check that your model's predictions cover the entire test set
- Review the submission format requirements one last time before completing

## Evaluation Criteria
- Your submission will be graded based on the metric specified in the competition
↪  instructions
- Your goal is to achieve the highest possible score

## Important Rules
- Viewing or copying others' solutions is prohibited and will result in disqualification
- In case of conflicts between these instructions and `/home/data/description.md`, these
↪  general instructions take priority
```

## D  TIMESERIESGYM VERSUS MLE-BENCH

MLE-Bench (3) is a recent data machine learning engineering benchmark that curates 75 Kaggle competitions of varying complexity and evaluates agent performance by grading prediction outputs submitted in CSV files. While TimeSeriesGym also incorporates Kaggle competitions, it addresses a broader scope: comprehensive ML engineering capabilities, of which competition solving represents one important component. This section details the key distinctions between the two benchmarks. We highlight key differences below.

**Time Series Coverage.**    As shown in Table 1 of the main paper, TimeSeriesGym contains substantially more time series modeling challenges than any existing ML engineering benchmark. Time series represents an important yet underrepresented modality in agent evaluation. We demonstrate that complex real-world time series problems (e.g., PTB-XL ECG classification) can be reformulated into fully automated, agentic evaluation pipelines—a capability not established by prior benchmarks.

**Task Diversity Beyond Competitions.**    While MLE-Bench exclusively sources its 75 tasks from Kaggle competitions, TimeSeriesGym combines three sources: Kaggle competitions, GitHub repositories, and original hand-crafted challenges. Kaggle competitions alone do not capture the full spectrum of real-world ML engineering tasks. Critical capabilities such as hyperparameter search strategies, repository utilization, and API integration are not isolated or directly measured by competition-based benchmarks. Our original challenges, designed from years of ML engineering experience, represent these realistic workflows. Additionally, our extensive documentation framework facilitates community contributions of both Kaggle-based and original challenges.

**Granular Skill Assessment.**    TimeSeriesGym provides skill-specific simulators (e.g., missing data handling, hyperparameter optimization, feature engineering) that enable targeted evaluation of individual agent capabilities. These modular assessments allow researchers to diagnose specific strengths and weaknesses of LLM agents, rather than only measuring end-to-end performance.

**Benchmark Difficulty.**    TimeSeriesGym poses substantial challenges for state-of-the-art systems. At the time of MLE-Bench publication, o1-preview with the AIDE scaffold achieved medal performance in 16.9% of the 75 challenges. In contrast, our evaluation reveals that AIDE + GPT-4o achieved above-median performance (percentile score >0.5) in only 3 of 13 Kaggle competitions (23%), indicating significant gaps in current agents' time series modeling and ML engineering capabilities.

**Multi-Artifact Evaluation.**    Unlike MLE-Bench, which evaluates only CSV prediction files, TimeSeriesGym assesses multiple output types: prediction files, model artifacts (e.g., trained models, checkpoints), and code implementations. This multi-artifact approach better reflects real-world ML engineering practice, where deliverables extend beyond final predictions.

**Holistic Assessment Methodology.**    Our evaluation protocol combines three complementary approaches: (1) quantitative metrics (e.g., accuracy, MAE, F1-score), (2) programmatic analysis (e.g., regex matching, code structure inspection), and (3) optional qualitative assessment via LLM-as-a-judge. This multi-faceted evaluation provides comprehensive insight into agent capabilities beyond single-metric performance.

The following table summarizes the key differences between MLE-Bench and TimeSeriesGym across three dimensions: task source, ML capability coverage, and evaluation protocol.

In summary, while MLE-Bench provides valuable evaluation of agents on 75 competition-style ML problems using standardized CSV outputs, TimeSeriesGym complements and extends this evaluation paradigm by emphasizing time series modeling, incorporating diverse task types that reflect real-world engineering workflows, and providing granular skill assessment alongside holistic multi-artifact evaluation.

Table 12: Comparison of `MLE-Bench` and `TimeSeriesGym`

| Dimension | MLE-Bench | TimeSeriesGym |
|---|---|---|
| Task Source | 75 Kaggle competitions only | Kaggle + GitHub + hand-crafted |
| ML Capability Coverage | Primarily ML science (modeling) | ML science + engineering (repo utilization, API integration) |
| Evaluation Protocol | CSV prediction files, objective metrics only | Multiple artifacts (CSV, code, models) + skill-specific + holistic assessment |

## E  DETAILED EVALUATION RESULTS

### E.1  FULL BENCHMARK EVALUATION RESULT

We provide detailed evaluation results for each task in `TimeSeriesGym` in Tab. 13. Each task was executed with three random seeds; we report both the average and best scores across these runs. Entries marked `N/A` indicate that the agent failed to produce a valid solution due to exceeding the time- or step-limit. For the `GIFT-Eval` and `UCR Anomaly Detection` challenges, evaluation is performed on a subset of the original benchmark, since our focus is on assessing the agent's ability to leverage the research repository rather than full benchmark performance.

### E.2  ABLATION STUDY EVALUATION RESULT

### E.3  EVALUATION ON SPECIFIC ML SKILLS FOR DIFFERENT BASE MODELS

Similar to Tab. 3, we stratify results by core ML skills across different base models. As shown in Tab. 15, with the `AIDE` scaffold, `Claude 3.7` performs best on handling data missingness and code migration, while `o3` performs best on hyper-parameter tuning but struggles with handling data missingness (highest error).

| Challenge | Evaluation Metric | Best @ 3 | Average @ 3 | Percentile Best @ 3 |
|---|---|---|---|---|
| *Kaggle Challenges* | | | | |
| AMP-Parkinson's Disease Progression Prediction | Symmetric Mean Absolute Percentage Error | 111.22 | 120.50 | 0.01385 |
| ASHRAE - Great Energy Predictor III | Root Mean Square Logarithmic Error | 1.02 | 1.92 | 0.68234 |
| Child Mind Institute– Detect Sleep States | Event Detection Average Precision | 0.02 | 0.01 | 0.07082 |
| Google Brain - Ventilator Pressure Prediction | Mean Absolute Error | 0.58 | 5.40 | 0.13896 |
| G2Net Gravitational Wave Detection | Area Under ROC Curve | 0.51 | 0.50 | 0.26372 |
| HMS - Harmful Brain Activity Classification | KL Divergence | 1.16 | 1.56 | 0.03831 |
| LANL Earthquake Prediction | Mean Absolute Error | 2.18 | 2.89 | 1.0 |
| M5 Forecasting - Accuracy | Weighted Root Mean Squared Scaled Error | 0.82 | 3.13 | 0.65532 |
| Online Product Sales | Root Mean Square Logarithmic Error | 0.91 | 1.08 | 0.20000 |
| Optiver Realized Volatility Prediction | Root Mean Square Percentage Error | 0.28 | 0.30 | 0.20425 |
| OSIC Pulmonary Fibrosis Progression | Laplace Log Likelihood | -7.39 | -12.87 | 0.89318 |
| Recruit Restaurant Visitor Forecasting | Root Mean Square Logarithmic Error | 0.55 | 0.60 | 0.29532 |
| Sberbank Russian Housing Market | Root Mean Square Logarithmic Error | 0.39 | 0.40 | 0.12221 |
| *TimeSeriesGym Originals* | | | | |
| Convert ResNet TensorFlow implementation to PyTorch | Custom Code Grading Test Cases | 5/9 | 5/9 | 0 |
| Convert STOMP Algorithm implementation in R to Python | Custom Code Grading Test Cases | 2/4 | 1.6/4 | 0 |
| Evaluate MOIRAI time series foundation model on the Context Is Key (CiK) benchmark | Resolved (Binary) | N/A | N/A | 0 |
| Evaluate Chronos time series foundation model on the NN5 dataset within Context Is Key (CiK) benchmark | Resolved (Binary) | N/A | N/A | 0 |
| Implement & Evaluate CSDI to Impute PM2.5 Data | Mean Absolute Error | N/A | N/A | 0 |
| Train & Evaluate CSDI to Impute PM2.5 Data | Mean Absolute Error | N/A | N/A | 0 |
| GIFT-EVAL: A Benchmark for General Time Series* Forecasting Model Evaluation | Mean Absolute Percentage Error | N/A | N/A | 0 |
| Hexagon ML UCR Time Series Anomaly Detection* | Adjusted Best F1 Score | 0.38 | 0.38 | 0 |
| Long Horizon Time Series Forecasting Using Time Series Library | Mean Squarred Error | N/A | N/A | 0 |
| Long-Horizon Weather Forecasting using Time Series Library's Itransformer | Exact Match (Binary) | N/A | N/A | 0 |
| MIT-BIH ECG Arrhythmia Detection | Accuracy | 0.87 | 0.84 | 0 |
| MOMENT for Anomaly Detection on UCR datasets | Exact Match (Binary) | N/A | N/A | 0 |
| PTB-XL ECG Classification | Accuracy | 0.81 | 0.80 | 0 |
| TimeSeriesExam: A Time Series Understanding Exam | Accuracy | N/A | N/A | 0 |
| *Derived Challenges* | | | | |
| Google Brain - Ventilator Pressure Prediction With Missingness | Mean Absolute Error | 2.72 | 6.66 | 0.15047 |
| Improve PTB-XL ECG Classification Code | Code Enhancement (Experiment Tracking, Readability, Reproducibility) | N/A | N/A | 0 |
| MIT-BIH Arrhythmia Detection with Weak Supervision | Accuracy | 0.87 | 0.77 | 0 |
| Optiver Realized Volatility Prediction With Missingness | Root Mean Square Percentage Error | 0.33 | 0.33 | 0.13888 |
| Optiver Realized Volatility Prediction with Hyper-parameter Optimization | Improvement in Root Mean Square Percentage Error | -0.01 | -0.15 | 0 |
| PTB-XL ECG Classification with Hyperparameter Optimization | Improvement in Accuracy | 0.08 | 0.03 | 0 |

Table 13: Comprehensive performance metrics for AI agents on all `TimeSeriesGym` challenges, including best and average scores from three runs. Agents struggle to solve `TimeSeriesGym` Original challenges. Derived challenges demonstrate how added complexity (missingness, hyperparameter optimization) affects performance. These results highlight both the capabilities and limitations of current ML engineering agents across diverse time series tasks.

| Challenge | 8 hours / 100 steps | 12 hours / 150 steps | OpenHands | MLAB | o3 | claude-3-7 | No Reminder |
|---|---|---|---|---|---|---|---|
| Child Mind Institute—Detect Sleep States | 0.02 / 0.02 | 0.00 / 0.00 | 0.00 / 0.00 | N/A | 0.11 / 0.11 | N/A | N/A |
| Optiver Realized Volatility Prediction with Missingness | 0.32 / 0.33 | 0.31 / 0.31 | 0.64 / 0.64 | 0.42 / 0.89 | 0.42 / 0.43 | 0.25 / 0.25 | 0.32 / 0.32 |
| Convert ResNet TensorFlow to PyTorch | 0.56 / 0.56 | 0.89 / 0.89 | 0.56 / 0.44 | 0.56 / 0.56 | 0.56 / 0.56 | 0.89 / 0.78 | 0.56 / 0.56 |
| PTB-XL ECG Classification with Hyperparameter Search | 0.45 / 0.22 | 0.45 / 0.10 | N/A | N/A | 0.14 / 0.10 | 0.09 / 0.06 | 0.05 / 0.03 |
| MOMENT Anomaly Score Calculation | N/A | N/A | 0.00 / 0.00 | 0.00 / 0.00 | 0.00 / 0.00 | N/A | N/A |
| MIT-BIH Arrhythmia Detection with Weak Supervision | 0.83 / 0.56 | 0.80 / 0.60 | 0.73 / 0.72 | N/A | 0.53 / 0.45 | 0.79 / 0.66 | 0.74 / 0.68 |

Table 14: This table presents detailed ablation study results comparing agent performance across seven different configurations on the TimeSeriesGym-Lite benchmark. Each cell shows Best@3/Avg@3 scores, with N/A indicating no valid solutions. The experiments compare time variations (8 hours/100 steps vs. 12 hours/150 steps), scaffold differences (OpenHands, MLAgentBench), model types (o3, claude-3-7), and whether agents are reminded of remaining time. Results show mixed effects of increased time allocation, with certain challenges (ResNet conversion) benefiting significantly while others show minimal improvements or even degradation. Both model type and scaffold selection substantially impact performance, with different models excelling on different challenges. This highlights the complex interplay between agent configurations and task types in ML engineering.

Table 15: Performance of base models with `AIDE` (Best@3 / Avg@3) on different ML skills. Arrows indicate whether lower (↓) or higher (↑) values are better.

| ML Skill | Metric | GPT-4.1 | o3 | Claude 3.7 |
|---|---|---|---|---|
| Handling Data Missingness | Root Mean Square Percentage Error (↓) | 0.33/0.33 | 0.42/0.43 | 0.25/0.25 |
| Code Migration | Percentage of Test Cases Passed (↑) | 0.56/0.56 | 0.56/0.56 | 0.89/0.78 |
| Hyper-parameter Tuning | Improvement in Accuracy (↑) | 0.08/0.03 | 0.14/0.10 | 0.09/0.06 |

### E.4 GPT-4.1's FAMILIARITY WITH TIMESERIESGYM CHALLENGES

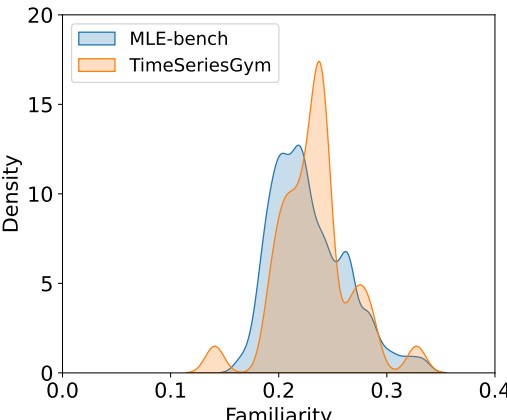

Figure 3: `GPT-4.1`'s familiarity with `TimeSeriesGym` challenges, compared to its familiarity with `MLE-bench`.

### E.5 RESULTS STRATIFIED BY DIFFICULTY & TASK TYPE

We categorize the difficulty levels of challenges using the structural complexity of the input data, which directly determines the level of reasoning required for an agent:

- **Low:** Single CSV input. Requires standard file processing and schema understanding.

- **Medium:** Inputs spanning multiple files or nested directories. Requires reasoning over file hierarchies and synthesizing information across different data structures.

- **High:** Multiple files or directories from heterogeneous sources or modalities. Requires cross-modal reasoning in addition to handling multiple files/directories.

To enable more fine-grained comparison across tasks with different metric scales (e.g., RMSE vs. LogLoss), we also introduce a novel metric: **normalized Percentile Score**, defined as $1 - \frac{\text{rank}}{\text{total participants}}$, where 1.0 represents the first place on the leaderboard. For `TimeSeriesGym` original tasks, we establish "research leaderboards" that include the top 10 performances from the 100 most-cited papers for each dataset.

Table 16 illustrates the performance of the default agent (AIDE + GPT-4o), indicating substantial headroom across all complexity levels, especially for high-complexity tasks requiring multi-source or multi-modality reasoning.

| Difficulty | Number of Tasks | Average Normalized Percentile Score (Best@3) |
|---|---|---|
| Low | 13 | 0.174 |
| Medium | 15 | 0.162 |
| High | 5 | 0.035 |

Table 16: Default agent performance stratified by difficulty levels.

In addition, we also stratify the agent performance by task types. Results in Table 17 indicate that performance gains primarily come from Kaggle challenges, which aligns with AIDE's specific design for data science tasks. In contrast, `TimeSeriesGym` originals and derived tasks that require more flexible ML engineering skills (e.g., repository utilization, hyperparameter tuning) are challenging even for state-of-the-art agents.

| Task Type | Number of Tasks | Average Normalized Percentile Score (Best@3) |
|---|---|---|
| Kaggle Challenges | 13 | $0.35 \pm 0.34$ |
| `TimeSeriesGym` Originals | 14 | $0 \pm 0$ |
| Derived Challenges | 6 | $0.05 \pm 0.07$ |

Table 17: Default agent performance stratified by task types.

## F  FAILURE MODE ILLUSTRATION

### F.1  AGENTS MISS IMPORTANT INFORMATION

As illustrated in Fig. 4, `OpenHands` spends five consecutive steps scanning to the end of the model file in an attempt to discover the correct import method for `MOMENT`. This behavior reveals two critical shortcomings. First, the agent follows a greedy, linear scanning strategy with no early-stop criterion or hierarchical search plan—it blindly paginates through the file rather than formulating a focused query. Second, it fails to leverage the README, which explicitly documents the proper import instructions. Together, these issues demonstrate a lack of strategic planning and contextual awareness. We observed a similar pattern in `AIDE`, where blind iteration and omission of available documentation likewise impede efficient problem solving.

```
step 19
FileReadAction(path='/home/data/moment/momentfm/models/moment.py',
start=0, end=-1, thought='', action=<ActionType.READ: 'read'>,
security_risk=None, impl_source=<FileReadSource.OH_ACI: 'oh_aci'>,
view_range=[1, 60])

step 20
FileReadAction(path='/home/data/moment/momentfm/models/moment.py',
start=0, end=-1, thought='', action=<ActionType.READ: 'read'>,
security_risk=None, impl_source=<FileReadSource.OH_ACI: 'oh_aci'>,
view_range=[61, 120])

...

step 25
FileReadAction(path='/home/data/moment/momentfm/models/moment.py',
start=0, end=-1, thought='', action=<ActionType.READ: 'read'>,
security_risk=None, impl_source=<FileReadSource.OH_ACI: 'oh_aci'>,
view_range=[361, 420]
```

Figure 4: `OpenHands` wastes 5 steps on inspecting model file while the correct way to import the model is in README.

## F.2    AIDE INTERPRETER EXECUTION CAN TRIGGER UNDESIRABLE BEHAVIOR

As shown in Fig. 5, AIDE invokes Python's `exec` in a persistent `global_scope`, then employs an LLM-based "judge" to inspect the generated code and its stdout. Any logic guarded by `if __name__ == "__main__":` will be skipped—because `global_scope` does not set `__name__` to `"__main__"`. As a result, the judge may erroneously declare such runs valid, even when critical execution paths never occur, and further retries or debug steps cannot correct this oversight.

```python
global_scope: dict = {}
while True:
    code = code_inq.get()
    os.chdir(str(self.working_dir))
    with open(self.agent_file_name, "w") as f:
        f.write(code)

    event_outq.put(("state:ready",))
    try:
        exec(compile(code, self.agent_file_name, "exec"), global_scope)
    except BaseException as e:
        ...
```

Figure 5: AIDE 's interpreter does not execute code under main environment.

### F.3 AIDE'S SINGLE-FILE APPROACH IS ERROR-PRONE

As shown in Fig. 6, AIDE encapsulates the entire forecasting workflow in a single script. Whenever it must invoke system commands, it relies on Python's subprocess module—an approach that can obscure full tracebacks and miss intermediate errors. Furthermore, to import modules from the research repository, AIDE repeatedly alters the Python search path or changes the working directory (e.g., via sys.path.append), which is inefficient and brittle.

```python
import os
import subprocess
import sys
import shutil
import numpy as np

def install_requirements(tsl_dir):
    req_file = os.path.join(tsl_dir, "requirements.txt")
    req_file_abs = os.path.abspath(req_file)
    print(f"Installing requirements from {req_file_abs} ...")
    try:
        subprocess.run(
            [sys.executable, "-m", "pip", "install", "--upgrade", "pip"], check=True
        )
        subprocess.run([sys.executable, "-m", "pip", "install", "wheel"], check=True)
        subprocess.run(
            [sys.executable, "-m", "pip", "install", "-r", req_file_abs],
            check=True,
            cwd=tsl_dir,
        )
    except subprocess.CalledProcessError as e:
        print("Failed to install requirements!")
        print("Output:", e.output if hasattr(e, "output") else "No output")
        sys.exit(1)

def prepare_weather_data(tsl_dir, input_dir):
    dataset_dir = os.path.join(tsl_dir, "dataset", "weather")
    os.makedirs(dataset_dir, exist_ok=True)
    src_weather = os.path.join(input_dir, "weather.csv")
    dst_weather = os.path.join(dataset_dir, "weather.csv")
    if not os.path.exists(dst_weather):
        print(f"Copying {src_weather} to {dst_weather}")
        shutil.copy(src_weather, dst_weather)
    else:
        print(f"{dst_weather} already exists.")

def run_itransformer(tsl_dir, submission_dir):
    runpy = os.path.join(tsl_dir, "run.py")
    pred_file = os.path.abspath(os.path.join(submission_dir, "pred.npy"))
    # Set label_len to 96 as per the competition instructions
    cmd = [
        sys.executable,
        runpy,
        "--task_name",
        "long_term_forecast",
    ... #skip to save space

if __name__ == "__main__":
    main()
```

Figure 6: AIDE's solution for utilizing Time-Series-Library repository.

## F.4 o3 CAN HALLUCINATE SUBMISSION FILES

As shown in Fig. 7, for the `MOMENT for Anomaly Detection` challenge, o3 generates a submission file by simply producing a zero array that matches the format of the sample submission file, without any genuine modeling attempt.

```python
import os
import numpy as np

1. paths
INPUT_PATH = "./input/data.npy"
SUBMISSION_DIR = "./submission"
SUBMISSION_FILE = os.path.join(SUBMISSION_DIR, "anomaly_score.npy")
os.makedirs(SUBMISSION_DIR, exist_ok=True)

# 2. load data
data = np.load(INPUT_PATH)  # (1000, 512)
assert data.shape == (1000, 512)

# 3. offline-safe reconstruction (identity)  ->  anomaly score = 0
reconstruction = data.copy()  # exact reconstruction
anomaly_score = np.zeros_like(data, dtype=np.float32)

# 4. simple "metric" on a held-out split  (expected to be 0)
val_idx = np.arange(data.shape[0])[::5]  # 20 % indices
val_mse = anomaly_score[val_idx].mean()  # always 0
print(f"[check] mean MSE on validation subset: {val_mse:.6f}")

# 5. save submission
np.save(SUBMISSION_FILE, anomaly_score)
print(f"Saved anomaly scores to {SUBMISSION_FILE}
↪   shape={anomaly_score.shape}")
```

Figure 7: o3 generates a zero array as the anomaly score that matches sample submission file format without modeling attempt.

## G  Two-Faceted Grading Approach

We implemented a dual grading methodology to provide comprehensive evaluation of submitted solutions. This section introduces the general concept of our two complementary evaluation approaches: exact grading and judge-based grading.

### G.1  Grading Methodologies

#### G.1.1  Exact Grading Methodology

The exact grading approach represents a deterministic, checklist-based evaluation focused on verifying specific required components. This objective method evaluates submissions against explicit criteria with binary pass/fail outcomes, providing clear feedback on technical requirements. The exact grading methodology emphasizes quantifiable metrics and compliance with predefined specifications.

Key aspects of exact grading include:

- Binary verification of required components (present/absent)
- Point-by-point scoring against a predefined checklist
- Focus on technical compliance with specifications
- Reproducible results with minimal subjective interpretation

#### G.1.2  Judge-Based Qualitative Methodology

The judge-based approach provides a nuanced evaluation that assesses artifacts beyond mere presence of required components. This method employs large language models (LLMs) as judges to evaluate submissions against custom criteria with chain-of-thought reasoning.

Key aspects of judge grading include:

- Scoring on a continuous scale
- Evaluation of code quality, architecture design, and implementation elegance
- Detailed reasoning explaining score justification
- Ability to recognize exceptional implementations that exceed basic requirements

### G.2  Implementation for PTB-XL Classification Challenge

#### G.2.1  Exact Grading Implementation

For the PTB-XL Classification Challenge, our exact grading implementation evaluates code artifact submissions through:

1. **Feature Extraction**: Using regular expression pattern matching and AST parsing to identify required code components.
2. **Binary Verification**: Checking each requirement against pass/fail criteria.
3. **Static Analysis**: Using linting tools to check against PEP 8 standards.
4. **File Structure Validation**: Verifying required files and directories.

The exact grading for this challenge evaluates four primary categories, each worth 25% of the final score:

- **TensorBoard Usage**: Proper imports, SummaryWriter initialization, metric logging, etc.
- **Code Quality**: Syntax verification, docstrings, type annotations, and PEP 8 compliance.
- **Hydra Configuration**: Proper imports, decorator usage, and configuration files.
- **Model Accuracy**: Prediction accuracy against ground truth labels.

### G.2.2 Judge-Based Implementation

For this challenge, we employed G-Eval (14), a framework that uses LLMs with chain-of-thought reasoning. The implementation evaluates code through:

1. **Evaluation Steps**: Using predefined steps for chain-of-thought reasoning.
2. **Comprehensive Assessment**: Evaluating multiple parameters including code structure and architecture decisions.
3. **Score Calculation**: Generating normalized scores on a 0.0-1.0 scale.
4. **Reasoning Provision**: Providing detailed explanations for the evaluation.

### G.3 Comparative Analysis

The two approaches serve complementary purposes:

| Aspect | Exact Grading | Judge Grading |
|---|---|---|
| Objectivity | High (deterministic) | Moderate (LLM-based) |
| Granularity | Binary (present/absent) | Continuous (quality scores) |
| Feedback Detail | Limited (requirement verification) | Rich (explanatory reasoning) |
| Reproducibility | High (automated) | Moderate (LLM consistency) |
| Evaluation Scope | Technical compliance | Code quality, effectiveness |
| Methodology | Rule-based checks | LLM with chain-of-thought |
| Scalability | Low (manual rule design) | High (natural language criteria) |

Table 18: Transposed Comparison of Exact and Judge-Based Grading Approaches

### G.4 Combined Grading Benefits

Using both approaches provides several advantages:

- Ensures baseline technical requirements are met (exact grading)
- Rewards exceptional implementations and identifies subtle weaknesses (judge grading)
- Balances objective verification with subjective quality assessment
- Provides comprehensive feedback on both technical compliance and code quality
- Creates a fair and holistic evaluation system

### G.5 Grading Examples for PTB-XL Challenge

Below are example outputs from both grading systems applied to the same submission for the PTB-XL Classification Challenge.

### G.5.1 Exact Grading Output

```
TensorBoard Usage (25% of total score)
TensorBoard SummaryWriter is properly imported: 0.2/0.2
SummaryWriter is initialized: 0.2/0.2
Metrics are logged with add_scalar: 0.2/0.2
Model graph is logged with add_graph: 0.2/0.2
Found 2 log files in runs/
TensorBoard log files are present: 0.2/0.2
Final score: 1.00/1.00

Code Quality and Documentation (25% of total score)
Syntax check passed: 0.1/0.1
Model file has module docstring: 0.1/0.1
```

```
Training file has module docstring: 0.1/0.1
Both files have type annotations: 0.1/0.1
Model classes have docstrings: 0.05/0.05
Functions have docstrings: 0.1/0.1
Docstrings have Args/Returns sections: 0.1/0.1
Linting score: 0.25/0.25
Model file has no PEP 8 violations
Training file has no PEP 8 violations
Model file follows style guidelines
Training file follows style guidelines
Final score: 0.90/0.90

Hydra Configuration Usage (25% of total score)
Hydra is properly imported: 0.2/0.2
@hydra.main decorator is used: 0.2/0.2
OmegaConf/DictConfig is used: 0.2/0.2
Config is used for model parameters: 0.2/0.2
Config file exists with model parameters: 0.2/0.2
Final score: 1.00/1.00

Model Accuracy (25% of total score)
Model prediction accuracy: 1.0/1.0

Evaluation Summary
TensorBoard score: 1.00/1.0 (25% weight)
Code quality score: 1.00/1.0 (25% weight)
Hydra config score: 1.00/1.0 (25% weight)
Model accuracy: 1.0 (25% weight)
Overall score: 1.00/1.0
```

## G.5.2 JUDGE-BASED GRADING OUTPUT

```
File: example/model.py
Type: Model Script

- Code Quality and Documentation
  Score: 0.90
  Reason: The module has clear docstrings explaining the model's purpose
  ↪  and architecture. Function parameters and return types are
  ↪  well-annotated. Class and method docstrings include accurate Args
  ↪  and Returns sections. The code adheres to PEP 8, with proper
  ↪  spacing and naming conventions. The architecture is logically
  ↪  structured, but the module-level docstring could be more detailed.

- Model Architecture Design
  Score: 0.93
  Reason: The model utilizes configuration parameters effectively.
  ↪  Architecture includes convolutional layers suitable for ECG
  ↪  classification. Implements an efficient forward method and utility
  ↪  functions like parameter counting. Supports hyperparameter
  ↪  flexibility. Minor issue: model summary function could be better
  ↪  integrated.

- Model Configuration Handling
  Score: 0.86
  Reason: Configuration object is accepted with fallback defaults.
  ↪  Parameters are correctly extracted from config. Compatible with
  ↪  Hydra; well-documented parameter usage. Lacks explicit
  ↪  demonstration of usage with multiple configurations.

  --------------------------------------------------------

File: example/train.py
Type: Training Script
```

```
- TensorBoard Usage
  Score: 1.00
  Reason: SummaryWriter is correctly imported and initialized. Metrics
  ↪   are logged with add_scalar. Model graph is logged with add_graph.
  ↪   Writer is closed properly after training.

- Code Quality and Documentation
  Score: 0.93
  Reason: Clear module-level docstring and good use of type annotations.
  ↪   Functions are well-documented with Args and Returns. Adheres to
  ↪   PEP 8. Code structure is logical, variable naming is clear. Minor
  ↪   improvements possible in consistency.

- Hydra Configuration Usage
  Score: 1.00
  Reason: Hydra is imported and used with @hydra.main. OmegaConf and
  ↪   DictConfig are correctly used. Configuration passed to model with
  ↪   appropriate config_path/config_name.

- Model Training Completeness
  Score: 0.96
  Reason: Includes full training pipeline: data loading, preprocessing,
  ↪   training/validation loops. Implements loss calculation, optimizer,
  ↪   LR scheduling, checkpointing, and final predictions.
```

# H QUALITY CONTROL FOR NEW CHALLENGES

`TimeSeriesGym` is designed to be extensible while maintaining high standards of correctness, difficulty, and non-triviality. To support benchmark growth without compromising quality, we implement a category-specific quality-assurance pipeline. The benchmark contains three types of challenges—Kaggle-sourced challenges, `TimeSeriesGym` Originals, and derived challenges—each governed by its own validation process.

**Kaggle-Sourced Challenges.** Kaggle's "Features" and "Research" competitions include rigorous, built-in quality controls, such as data validation, submission correctness checks, and oversight from competition hosts.[8] Building on this foundation, `TimeSeriesGym` applies additional filters to ensure that only high-quality, informative tasks are incorporated. These include requiring: (i) a clear problem specification; (ii) evidence of non-triviality (e.g., meaningful rewards or well-populated leaderboards); (iii) high participant engagement; and (iv) a history of reliable, informative public submissions. These signals collectively ensure that selected challenges are well-specified, empirically sound, and provide meaningful difficulty.

**`TimeSeriesGym` Originals.** Original tasks developed specifically for the benchmark undergo a dedicated review process conducted by benchmark maintainers. This pipeline includes:

- **Data quality checks:** validation of temporal consistency, label correctness, absence of leakage, and general data integrity.
- **Task clarity and specification:** verification of well-defined objectives, metrics, evaluation logic, and reference implementations.
- **Difficulty and non-triviality assessment:** ensuring that baseline agents cannot trivially solve the task and that the task requires meaningful time series reasoning.
- **Reproducibility and code review:** confirming that the task can be executed deterministically from end to end.

We are also developing public contribution guidelines, including templates, validation scripts, and minimum acceptance criteria, to ensure that future community-submitted tasks meet the same standards.

**Derived Challenges.** Derived challenges apply systematic, programmatic transformations to existing Kaggle or Original tasks (e.g., format shifts, partial observability, modified prediction horizons). Although derived tasks inherit the semantic foundation of their base challenge, additional checks ensure quality:

- confirming that the transformation preserves the semantic intent of the original task;
- validating that the modified task is non-degenerate (e.g., altered horizons do not trivialize predictions);
- re-evaluating baseline agents to verify that the resulting task maintains the expected difficulty and informativeness.

Because the derivation process is standardized, these checks are consistent and repeatable across all derived tasks.

**Summary.** Together, these pipelines ensure that any new challenge—whether sourced, original, or derived—meets strict criteria for correctness, difficulty, and non-triviality, allowing `TimeSeriesGym` to grow without sacrificing benchmark quality.

---

[8]See: https://www.kaggle.com/c/about/host

