# OpenReview forum: "TimeSeriesGym: A Scalable Benchmark for (Time Series) Machine Learning Engineering Agents"
_ICLR.cc/2026/Conference — Submitted to ICLR 2026_

### Official Review · Reviewer_E1hF · 2025-10-27

**Soundness:** 2
**Presentation:** 2
**Contribution:** 1
**Rating:** 2
**Confidence:** 3

**Summary:**

This paper introduces an open-source benchmark TimeSeriesGym for MLE agents evaluation, mostly focused on time series data. It contains 33 challenges and evaluated 3 scaffolds of agents with each scaffold evaluated on up to 3 base models. The framework supports multimodal output evaluation, specific skill evaluation and holistic evaluation(quantitative & qualitative).

**Strengths:**

This benchmark fills a gap on time series ML pipeline among existing ML benchmarks.

The proposed benchmark offers specific skills evaluation and hybrid scoring of qualitative + quantitive evaluation.

**Weaknesses:**

This paper emphasizes benchmark at scale which would require limited human efforts but there is no discussion on how exactly new challenges should be adapted to the time series gym. It's mentioned that new challenge can be added within two hours, but how is the quality of the generated new challenges? In addition, two hours time is probably also enough to set up a specialized evaluation for a challenge without the need to integrate it to the current evaluation framework. The evidence for claimed scalability is limited, at least not well-presented.

It feels like this paper is mostly doing the same thing as MLE-bench but with specifically curated challenges focusing on time series and additionally took inspiration from other benchmarks and combined their advantages such as multimodal, skill-based, and holistic evaluation.

The experiment findings are also very predictable. Existing agents don't have the full capability to automate complete ML pipeline and highly likely the reasoning model produces more reasonable solutions. It's hard to find new takeaways or interesting insights from this paper.

Some experiment settings are not meaningful. For example, providing remaining time reminders to agents vs. without time reminders. I find it hard to see the value in this experiment.

The evaluation is mostly limited to gpt 4.1 model, and there is no open source models being evaluated at all.

Some challenges feel very synthetic. For example, implement MOMENT for anomaly detection task. But why do we need to integrate such task into this benchmark at all given that the code to solve this task is already available in MOMENT official github repo.

**Questions:**

The paper lacks clarity. It's not clear how exactly the challenges are defined, are they all manually crafted based on Kaggle and Github Repos? Line 155-163 shows what each challenge is made or but is it manual effort to create these information based on the dataset or repository?

In line 315-316, it's mentioned that "simply loading and re-saving the provided sample submission file without any model inference or data processing is deemed unreasonable". How is this checked? Is it through llm-as-a-judge evaluation or manual inspection?

In line 228-231, the paper mentions that it can grade diverse artifacts generated throughout the MLE life cycle but how? I don't think this is explained. Rather than spending almost a whole page on limitation/future work/open questions, I would suggest spending more text to improve the clarity of the paper and explaining details of this benchmark framework in a thorough manner.

---

> ### Author Response · Authors · 2025-11-25
> **Author Response (1/5)**
>
> Dear Reviewer E1hF,
>
> We really appreciate your valuable comments and time. We value your feedback, and have spent significant time reflecting and acting on your comments. We are pleased to learn that you found our work to fill a significant gap in benchmarking the ability of agent of (time series) ML problems. We are also glad that you found skill-specific evaluation and hybrid scoring valuable.
>
> Please find our our response to your comments below.
>
> > W1. This paper emphasizes benchmark at scale which would require limited human efforts but there is no discussion on how exactly new challenges should be adapted to the time series gym. It's mentioned that new challenge can be added within two hours, but how is the quality of the generated new challenges? In addition, two hours time is probably also enough to set up a specialized evaluation for a challenge without the need to integrate it to the current evaluation framework. The evidence for claimed scalability is limited, at least not well-presented.
>
> We really appreciate this comment, and we agree that our presentation can be improved to better highlight the numerous mechanisms that make `TimeSeriesGym` scalable.
>
> We believe that the following design choices make `TimeSeriesGym` scalable by design:
>
> 1. **Scaling skill-specific competitions:** We provide specialized tools to create skill-specific challenges (e.g., simulating missing data) and evaluate them. These skill simulators can be used with any number of "base" competitions to generate a large number of diverse competitions which test for specific skills.
> 2. **Scaling agent outputs:** Our benchmarks provides tools to evaluate a large number of agent outputs, including prediction files, model artifacts, and code.
> 3. **Scaling agentic scaffolds:** Unlike existing benchmarks such as `MLGym`, `TimeSeriesGym` is agnostic to specific agent implementations. It is designed in a way that makes it easy to add new agent implementations.
> 4. **Data sources:** Kaggle comprises of an ever growing number of professional challenges, and many ML practitioners also have proprietary datasets on which they want to evaluate agents. Our benchmark is designed in a way to enable researchers to easily add and evaluate new tasks, whether Kaggle-style or not, as and when such challenges appear (see the following paragraph).
>
> Additionally, we believe that the following properties make `TimeSeriesGym` scalable in practice:
>
> 1. **Detailed documentation to add new challenges:** Our [detailed documentation](https://anonymous.4open.science/r/TimeSeriesGym-9CF6/documentation/) has allowed members of our community, but outside of our research team, to contribute a new challenge within 2 hours. Our publicly released anonymous code, had detailed step-by-step documentation to add new challenges and agentic scaffolds. While this evidence is anecdotal, nevertheless, we believe that it supports the scalability claim.
>
> We believe that these design decisions significantly lower the barrier and reduces the cost and implementation effort for reproducible agent evaluation, especially for practitioners who cannot release their data publicly. We will revise the manuscript to present it more clearly.

---

> ### Author Response · Authors · 2025-11-25
> **Author Response (2/5)**
>
> > W2. It feels like this paper is mostly doing the same thing as MLE-bench but with specifically curated challenges focusing on time series and additionally took inspiration from other benchmarks and combined their advantages such as multimodal, skill-based, and holistic evaluation.
>
> We are sorry that you feel this way, and hope that the following paragraphs change your mind in this regard.
>
> `MLE-Bench` is a well-designed benchmark, which primarily evaluates LLM agents on their ability to solve text and image Kaggle challenges. While `TimeSeriesGym` also sources challenges from Kaggle, it is designed to evaluate LLM agents on their ability to do ML engineering, of which solving Kaggle challenges is a small (but important) part. `TimeSeriesGym` has several important distinctions from `MLE-Bench`:
>
> 1. **Coverage of Time Series Tasks:** As shown in Table 1, `TimeSeriesGym` has far more challenges that require modeling time series data compared to any ML engineering benchmark. Time series is an important and underrepresented modality. We show that many real-world time series problems (e.g., PTB-XL ECG classification) can be reformulated into a fully automated, agentic evaluation pipeline, which is not demonstrated by prior benchmarks.
> 2. **Original challenges:** `MLE-Bench` exclusively sources challenges from Kaggle. Our extensive (and growing) documentation makes it easier for the community to add new Kaggle-based challenges easier. But Kaggle challenges alone do not reflect real-world (time series) machine learning engineering problems, such as hyperparameter search, repository utilization, and API integration capabilities that existing benchmarks, including `MLE-Bench`, do not isolate or measure directly. `TimeSeriesGym` includes original challenges, which represent realistic ML engineering problems motivated by years of experience of the research team.
> 3. **Broader ML capability coverage:** `TimeSeriesGym` has skill-simulators (e.g., data missingness handling, hyperparameter search) that none of the other benchmarks have, which enable researchers to evaluate specific capabilities of LLM agents.
> 4. **Significantly Harder:** `TimeSeriesGym` is challenging for state-of-the-art LLMs and agentic scaffolds. Meanwhile, existing benchmarks such as `MLE-Bench` are much easier for these agents. Note that, at the time of writing the paper, `o1-preview` with the AIDE scaffold achieved medal in 16.9% of the challenges. Meanwhile, our updated results reveal that the default setup (AIDE + GPT-4o) achieved a percentile score > 0.5 (above median) in only 3 out of 13 Kaggle competitions, which reveal a significant gap in the time series modeling and ML engineering capabilities of state-of-the-art agents.
> 5. **Multimodal outputs:** `MLE-Bench` can only evaluate CSV prediction files, whereas `TimeSeriesGym` can also evaluate model artifacts and code. These modalities are common in ML engineering practice.
> 6. **Holistic evaluation:** Our evaluation methodology deliberately combines multiple assessment approaches: quantitative metrics (e.g., accuracy, mean absolute error), programmatic analysis (e.g., regular expression matching, code inspection), and optional qualitative evaluation (LLM-as-a-judge). Each challenge in `TimeSeriesGym` is evaluated using quantitative metrics and _optionally_ subjective metrics (LLM judge).
>
> We are revising the manuscript to highlight these distinctions more clearly.

---

> ### Author Response · Authors · 2025-11-25
> **Author Response (3/5)**
>
> > W3. The experiment findings are also very predictable. Existing agents don't have the full capability to automate complete ML pipeline and highly likely the reasoning model produces more reasonable solutions. It's hard to find new takeaways or interesting insights from this paper.
>
> Thank you for this comment! We agree that some high-level findings may appear intuitive. However, the goal of our evaluation is not to produce surprising results, but to provide **fine-grained diagnostic insights into how agents perform on realistic ML engineering tasks and how they can improve**. As detailed in Appendix D, we identify several concrete and recurring failure modes, including:
>
> - **Loss of critical information** in long-context, multi-turn workflows.
> - **Scaffold-specific limitations**, such as AIDE's single-file strategy that fail on repository-level tasks.
> - **Hallucination under complex repository navigation**, where agents fabricate outputs instead of navigating through hierarchical directories and grounding in the codebase.
>
> We believe that these fine-grained analyses highlight the specific bottlenecks in current agent design. We believe these insights are valuable for the community, as they point to concrete, actionable directions for developing more reliable agentic systems for ML engineering.
>
> > W4. Some experiment settings are not meaningful. For example, providing remaining time reminders to agents vs. without time reminders. I find it hard to see the value in this experiment.
>
> We humbly disagree with this statement. As we motivate this experiment in Line 356:
>
> > We suspected that agents do not improve with more time because they do not use it well. To test this idea, we compared two settings: the default approach of reminding the agent about remaining time (and steps) before each step, versus removing these reminders completely. Surprisingly, we did not find significant differences between these settings. In fact, agents without time reminders produced more reasonable submissions. This may suggest that agents do not use their time wisely– they tend to rush toward solutions instead of carefully exploring promising options, especially towards the end of the experiment. This raises important research questions about how to design agents that use their time and resources more strategically.
>
> These experiments investigate a **critical yet underexplored factor** in agentic systems: **agents must operate within finite execution windows, and their behavior can change under time pressure**. This experiment aims to evaluate whether agents can strategically adapt their resource allocation strategy on the fly when given reminders.
>
> As shown in Table 2, agents with reminders produced more valid but fewer reasonable submissions, revealing that reminders may induce "rushed" behavior rather than strategic planning. This highlights an important issue: current agents do not manage their time budget effectively, even when explicitly informed about it. We believe this finding is meaningful as it highlights a concrete and actionable need for agents that can utilize their time and resources more strategically.
>
> That said, we agree that the rationale behind these experiments can be presented more clearly. We are making this change in the revised manuscript.

---

> ### Author Response · Authors · 2025-11-25
> **Author Response (4/5)**
>
> > W5. The evaluation is mostly limited to gpt 4.1 model, and there is no open source models being evaluated at all.
>
> Thank you for this suggestion! We agree that evaluating open-source models is essential for a comprehensive benchmark. We have updated Table 2 to include results from DeepSeek V3.2, evaluating both its thinking (reasoner) and chat (chat) modes on `TimeSeriesGym-Lite`, and the results are shown below:
>
> | Scaffold | Model | Resources (hours/steps) | Valid Submission (%) | Reasonable Submission (%) |
> |---|---|---|---|---|
> | AIDE | DeepSeek V3.2-Reasoner | 4 / 50 | $11.1 \pm 9.6$ | $11.1 \pm 9.6$ |
> | AIDE | DeepSeek V3.2-Chat | 4 / 50 |  $16.7 \pm 0$ | $16.7 \pm 0$ |
>
> The results led to several notable observations:
>
> - DeepSeek models failed on 5 of the 6 `TimeSeriesGym-Lite` tasks, producing buggy code in all cases except the ResNet code-migration challenge.
> - On the ResNet code migration task, DeepSeek models performed strongly, passing 8/9 test cases, compared with an average of 5/8 for GPT-4 and o3, demonstrating strong code understanding and translation capabilities.
> - As AIDE leverages an LLM to provide feedback based on code execution results, buggy code sometimes receive misleading "successful" analyses due to hallucination. This highlights a key limitation of LLM-based feedback in ML engineering workflow: it may hallucinate and therefore hinder effective debugging of the agent.
>
> While AIDE performs well on standard Kaggle challenges (as shown in `MLE-Bench`), `TimeSeriesGym` reveals its limitations when tasks require more comprehensive and flexible ML capabilities. This underscores the value of our benchmark for evaluating agents in realistic ML engineering scenarios that go well beyond the typical Kaggle format.
>
> > W6. Some challenges feel very synthetic. For example, implement MOMENT for anomaly detection task. But why do we need to integrate such task into this benchmark at all given that the code to solve this task is already available in MOMENT official github repo.
>
> This [challenge](https://anonymous.4open.science/r/TimeSeriesGym-9CF6/timeseriesgym/competitions/moment-anomaly-detection-score/description.md) represents a particularly common ML engineering task: use a newly released state-of-the-art model (e.g., MOMENT) to solve a problem by consulting its code, READMEs and tutorials. Since real-world ML engineering work often involves configuring and integrating existing codebases, assessing this capability is essential for determining the practical usefulness of an agent and its potential to reduce the manual effort in end-to-end ML pipelines.
>
> We believe challenges such as these represent realistic problems that researchers spend a significant amount of time and effort on a daily basis, and must be a part of any benchmark which measures an LLM agent's ability to do ML engineering.
>
> We are now adding motivations behind these challenges to the appendix in the revised paper.

---

> ### Author Response · Authors · 2025-11-25
> **Author Response (5/5)**
>
> > Q1. The paper lacks clarity. It's not clear how exactly the challenges are defined, are they all manually crafted based on Kaggle and Github Repos? Line 155-163 shows what each challenge is made or but is it manual effort to create these information based on the dataset or repository?
>
> Thank you for the question, and we apologize for the lack of clarity. To answer your specific question: yes, all challenges are manually crafted. As outlined in Lines 155–168 and illustrated in Figure 1, we manually standardize diverse raw sources (e.g., Kaggle challenges, GitHub repositories) into a unified structure comprising three components: (1) source data, (2) task description, and (3) grading function. We also provide a step-by-step illustration of this construction procedure in our anonymous repository [documentation](https://anonymous.4open.science/r/TimeSeriesGym-9CF6/documentation/adding_new_challenges.md).
>
> > Q2. In line 315-316, it's mentioned that "simply loading and re-saving the provided sample submission file without any model inference or data processing is deemed unreasonable". How is this checked? Is it through llm-as-a-judge evaluation or manual inspection?
>
> For the results reported in the paper, we relied on manual inspection to accurately determine reasonable submissions and to avoid potential bias from LLM-as-a-judge.
>
> To enable scalable reproduction of the study, the public release will include a Regex-based verification script that automatically checks for genuine modeling attempts. While rule-based checking generally has limitations (e.g., syntactic rigidity), our approach is robust in practice as we control the Docker environment and know exactly which packages are pre-installed. Although agents can technically install external libraries, our heuristic explicitly covers the vast majority of standard ML frameworks. We view this as a deliberate trade-off that avoids the stochasticity of LLM judges while ensuring reproducibility at scale, and we will continue to refine the verification rules to ensure its consistency with human judgements.
>
> > Q3. In line 228-231, the paper mentions that it can grade diverse artifacts generated throughout the MLE life cycle but how? I don't think this is explained. Rather than spending almost a whole page on limitation/future work/open questions, I would suggest spending more text to improve the clarity of the paper and explaining details of this benchmark framework in a thorough manner.
>
> By "diverse artifacts", we refer to the framework's ability to evaluate various types of outputs produced across the ML engineering workflow (e.g., CSV predictions, trained models, generated code). This is enabled through task-specific grading functions provided in the environment, which parse the submitted file(s) and compute appropriate metrics, e.g., parsing submitted CSV files and computing accuracy against ground-truth labels, or loading model weights and verifying the inference outputs. The code is already publicly available and is the best source of truth. We have clarified this in the revised manuscript by expanding the explanation of the grading framework in the main text, and in the appendix. That said, **we believe that a good (benchmarking) paper should provide a comprehensive discussion of its limitations, open questions and avenues for future work.**
>
> _We appreciate your thoughtful feedback and have carefully reflected on your reviews, conducted further experiments, and we are also making substantial revisions to the paper. If our responses have addressed your concerns, we would appreciate if you would consider revising your evaluation. Thank you again for your time and careful review!_
>
> Best regards, \
> Authors of TimeSeriesGym

---

> > ### Comment · Reviewer_E1hF · 2025-11-26
> >
> > Thank you for the response!
> >
> > For w1: “scaling skill-specific competitions”, “scaling agent outputs”, “agent-agnostic design”, use of Kaggle/proprietary data do not answer my question of benchmark scalability.
> >
> > The only concrete support for the “2 hours per challenge” claim is an anecdote that an external contributor added one challenge in that time. This is not enough to substantiate a central scalability claim, especially when there is no analysis of variability across tasks, contributors, or difficulty.
> >
> > The question of quality control for new challenges remains unanswered: how are correctness, difficulty, and non-triviality validated when a new task is added, especially if the benchmark is meant to grow? Without a clear quality-assurance process, “easy to add more tasks” can also mean “easy to add low-quality or uninformative tasks.”
> >
> > LLMs are generally very sensitive to prompt. How do you ensure linguistic diversity rather than human bias from the contributor in formatting the prompt. The quality of new tasks will be questionable. How do agents perform under diverse formulations of the same task?
> >
> >
> > For Q2: This seems to be a huge flaw in the evaluation. regex based evaluation is very brittle. As also mentioned by reviewer c71P, the evaluation protocol is very unreliable. How are the research community expected to reproduce your human evaluation or replace with llm-as-judge evaluation. The success criteria for reasonable submission is too vague. "What counts
> > as a reasonable attempt varies by challenge type. For Kaggle challenges, we define it as scoring
> > above median on the competition’s public leaderboard. For the remaining challenges, we consider a
> > submission reasonable if it made a genuine6 modeling attempt rather than hallucinating an output
> > that matches the submission format. For example, simply loading and re-saving the provided sample
> > submission file without any model inference or data processing is deemed unreasonable."

---

> > > ### Author Response · Authors · 2025-12-02
> > > **Rebuttal Response (1/3)**
> > >
> > > Dear Reviewer E1hF,
> > >
> > > > For w1: “scaling skill-specific competitions”, “scaling agent outputs”, “agent-agnostic design”, use of Kaggle/proprietary data do not answer my question of benchmark scalability. The only concrete support for the “2 hours per challenge” claim is an anecdote that an external contributor added one challenge in that time. This is not enough to substantiate a central scalability claim, especially when there is no analysis of variability across tasks, contributors, or difficulty.
> > >
> > > Thank you for raising this important point. Our core claim is that TimeSeriesGym is scalable by design—that the framework, abstractions, and tooling make it feasible to add new ML-engineering challenges at low marginal cost. The anecdote about a contributor adding a challenge in ~2 hours is provided as a sanity check, not as the sole or primary evidence for scalability, and we agree it does not constitute a systematic analysis of variability across contributors or task types.
> > >
> > > A rigorous empirical study of challenge-creation time, stratified across contributors with different expertise levels, across task difficulties, and across challenge types, would indeed be a meaningful direction. However, such a study would require recruiting and compensating human participants, designing controlled task-creation protocols, and evaluating time-on-task and error rates, which is **beyond the scope of this work and more appropriately situated in an HCI-focused study**. We now explicitly acknowledge this in the paper.
> > >
> > > Instead, our **scalability claim is grounded in the architectural decisions and abstractions underlying `TimeSeriesGym`**:
> > >
> > > - **Detailed documentation to add new challenges.** We provide detailed, step-by-step guidelines and templates for adding new challenges, which reduce ambiguity and lower the barrier for new contributors.
> > >
> > > - **Scaling skill-specific competitions.** Our design allows a single real-world task (e.g., a Kaggle forecasting competition) to be decomposed into multiple focused challenges that target different ML engineering skills. This allows the benchmark to grow horizontally from a single source task.
> > >
> > > - **Scaling agent outputs.** TimeSeriesGym supports diverse agent artifacts such as code, models, rather than constraining tasks to submission.csv outputs, as in MLE-Bench. This broadens the space of addable challenges beyond Kaggle-only tasks.
> > >
> > > - **Agent-agnostic evaluation**. Unlike benchmarks tied to a single LLM agent (e.g., MLGym), our evaluation is fully agent-independent. This ensures that as new agents and scaffolds appear, new challenges remain valid without modification, supporting long-term extensibility.
> > >
> > > Together, these design principles mean that adding new tasks does not require bespoke infrastructure, custom evaluation logic, or agent-specific adaptations. This is the sense in which we argue `TimeSeriesGym` is a **scalable benchmark**.
> > >
> > > We are revising the paper to clarify (1) that the anecdote is supplementary, (2) what we mean by “scalable by design,” and (3) why a systematic human study is outside the paper’s scope.

---

> > > > ### Author Response · Authors · 2025-12-02
> > > > **Rebuttal Response (2/3)**
> > > >
> > > > > The question of quality control for new challenges remains unanswered: how are correctness, difficulty, and non-triviality validated when a new task is added, especially if the benchmark is meant to grow? Without a clear quality-assurance process, “easy to add more tasks” can also mean “easy to add low-quality or uninformative tasks.”
> > > >
> > > >
> > > > This is a great question! `TimeSeriesGym` includes three types of challenges-- Kaggle-sourced challenges, TimeSeriesGym Originals, and derived challenges-- and each category has its own quality-control pipeline. Below, we outline how correctness, difficulty, and non-triviality are validated for each type:
> > > >
> > > > **Kaggle challenges:** Kaggle’s Features and Research competitions come with rigorous built-in [quality controls](https://www.kaggle.com/c/about/host), including data validation, submission verification, and competition-host oversight. Building on this foundation, we apply additional filters to select only high-quality challenges, such as requiring a clear problem statement, meaningful rewards, high participant engagement, and a history of informative public leaderboards. These signals collectively ensure the tasks are non-trivial, well-specified, and empirically sound.
> > > >
> > > > **TimeSeriesGym Originals:** TimeSeriesGym Originals are challenges created specifically for the benchmark. These challenges (will) undergo a dedicated review process by the maintainers (i.e., us). This includes:
> > > > - Data quality checks: validation of data integrity, temporal coherence, labeling correctness, and absence of leakage.
> > > > - Task clarity: ensuring well-defined objectives, metrics, evaluation logic, and reference implementations.
> > > > - Difficulty and non-triviality assessment: verifying that baseline agents cannot trivially solve the task and that the task captures meaningful aspects of time series reasoning.
> > > > - Code review + reproducibility checks: confirming that the task can be run end-to-end deterministically.
> > > >
> > > > We are also developing public contribution guidelines, including templates, validation scripts, and minimal acceptance criteria, to ensure community-submitted challenges meet the same standard.
> > > >
> > > > **Derived challenges:** Derived challenges are systematic transformations of existing Kaggle or TimeSeriesGym-Original tasks (e.g., format shifts, partial observations, modified horizons). Their quality is tied to the base challenge, but we still enforce additional checks:
> > > >
> > > > - Ensuring the transformation preserves the semantic intent of the original task. This is true by design, since no skill simulators modify the intent of the challenge.
> > > > - Validating that the derived task is non-degenerate (e.g., altered horizons don’t trivialize the prediction).
> > > > - Re-running baseline agents to confirm expected difficulty and informativeness.
> > > >
> > > > Because the derivation process is programmatic and standardized, these checks are consistent and repeatable.
> > > >
> > > > > LLMs are generally very sensitive to prompt. How do you ensure linguistic diversity rather than human bias from the contributor in formatting the prompt. The quality of new tasks will be questionable. How do agents perform under diverse formulations of the same task?
> > > >
> > > > Thank you for raising this important point about prompt sensitivity in LLMs. However:
> > > >
> > > > **Our benchmark is not designed to evaluate LLMs.**
> > > >
> > > > The primary agents we evaluate—and design tasks for—are ML engineering agents, not general-purpose LLMs. Tasks are realistic operational workflows that human ML engineers solve in practice; the README for each challenge serves as the equivalent of normal task documentation, not a carefully engineered prompt.
> > > >
> > > > **We intentionally allow linguistic diversity in READMEs.**
> > > >
> > > > We do **not** standardize READMEs beyond basic clarity and correctness. Diverse tones, writing styles, and levels of explicitness are desirable because real-world documentation is heterogeneous. Rigid linguistic standardization would artificially sanitize the benchmark, making it less representative of real ML engineering settings.
> > > >
> > > > **Agents are evaluated on task performance, not prompt robustness.**
> > > >
> > > > Our goal is to measure whether software agents can correctly implement pipelines, interpret specifications, and solve realistic ML tasks—not whether LLMs are robust to paraphrase variation. As such:
> > > >
> > > > - The README is part of the challenge specification, not part of the evaluation axis.
> > > > - We expect agents to read arbitrary but clear documentation formats, as ML engineers would.
> > > > - Prompt variation does not affect task correctness because evaluation is fully automated and grounded in the underlying data and metrics.
> > > >
> > > > Our documentation already has guidelines with examples, metric definitions, and evaluation steps, but we intentionally do not homogenize the language of the tasks.

---

> > > > > ### Author Response · Authors · 2025-12-03
> > > > > **Rebuttal Response (3/3)**
> > > > >
> > > > > > For Q2: This seems to be a huge flaw in the evaluation. regex based evaluation is very brittle. As also mentioned by reviewer c71P, the evaluation protocol is very unreliable. How are the research community expected to reproduce your human evaluation or replace with llm-as-judge evaluation.
> > > > >
> > > > > Thank you for highlighting this issue. We agree that a purely regex-based approach would be brittle and that manual evaluation is not scalable or easily repeatable. To address this, we replaced the previous manual setup with a two-stage evaluation pipeline combining simple heuristics with an LLM-as-a-judge, both of which are fully specified in the paper.
> > > > >
> > > > > First, we use rule-based heuristics (regex patterns) only to flag clearly invalid behaviors—such as reading sample_submission.csv, producing constant/random predictions, or omitting any modeling steps—and to detect genuine modeling signals such as .fit() calls and ML-library imports. This stage is deterministic and transparent.
> > > > >
> > > > > Second, for borderline cases, we apply a DeepEval GEval LLM-as-judge with explicit criteria and a carefully designed rubric. The judge evaluates whether the submitted code constitutes a genuine modeling attempt—i.e., whether it performs data loading, processing, and model inference—scoring each submission as unreasonable (0) or reasonable (1).
> > > > >
> > > > > This combined heuristic + LLM-judge approach directly addresses the reproducibility concerns: both components are fully documented, easy to re-run, and allow the research community to reproduce our process or substitute another LLM judge while preserving the same criteria.

---

### Official Review · Reviewer_9ifC · 2025-10-31

**Soundness:** 3
**Presentation:** 3
**Contribution:** 3
**Rating:** 4
**Confidence:** 4

**Summary:**

This paper presents TimeSeriesGym, a benchmark and environment for evaluating agents that perform end-to-end machine learning work on time series problems. The benchmark aggregates a collection of challenges that span forecasting, classification, anomaly detection, data cleaning, hyperparameter tuning, and code migration. It evaluates not only final predictions but also multiple artifacts such as code, trained models, and submission files. It defines aggregate metrics for validity and reasonableness and reports results across different toolchains and foundation models. The paper argues that time series is an underrepresented setting for agent evaluation and that real engineering workflows require capabilities beyond model inference. The experimental section compares several agent frameworks under controlled resources and reports ablations on time budgets and guidance strategies. The authors claim that TimeSeriesGym is scalable, extensible, and better aligned with the realities of time series engineering than prior leaderboards that focus on a single metric or a single output file.

**Strengths:**

- Clear motivation that time series engineering requires more than single step prediction and that agents should be evaluated on multi stage workflows.

- The authors collected a large number of datasets, with broad task coverage and inclusion of multiple domains which improves the ecological validity of the benchmark.

- In this paper, multi artifact evaluation that considers predictions, code quality checks, and trained models which aligns the benchmark with real practice.

- Agent agnostic design that supports different frameworks and enables trajectory collection for future training.

- A documented process for adding new tasks which supports scalability and long term maintenance.

**Weaknesses:**

1. The paper mixes the benchmark, the task generation mechanism, the multi artifact scoring, the trajectory data loop, and the time series focus without a clear primary to secondary hierarchy and without a concise contribution figure. It does not decompose difficulty across different time series task families and it does not design difficulty along agent reasoning dimensions.

2. Due to compute constraints most experiments are conducted on the Lite set of six tasks. Although the authors state that these tasks cover key skills there is no statistical validation of domain diversity or difficulty distribution which can bias conclusions toward low cost scenarios.

3. The contribution is primarily engineering. The paper lacks a unifying methodological or theoretical insight. Although this is a benchmark paper it would benefit from a central methodological principle that guides design choices across sections.

4. The table presents results for 4/50→12/150 and for Step wise reminder vs No reminder, but it does not include significance testing.

5. The definitions of Valid submission and Reasonable submission appear in prose in the methods section without a unified symbolization or an explicit decision boundary. This reduces clarity and hurts reproducibility.

**Questions:**

1. Can you provide stratified results by task family with confidence intervals and tests for significance. This would help to understand which capabilities drive aggregate gains.

2. How do you ensure that the Lite subset is representative of the full benchmark. Please include quantitative evidence of coverage and difficulty distribution.

3. Can you formalize the validity and reasonableness criteria using symbols and thresholds and show calibration plots or decision curves.

4. What steps can be taken to reduce reliance on leaderboard medians or external competitions that may change over time.

---

> ### Author Response · Authors · 2025-11-26
> **Author Response (1/5)**
>
> Dear Reviewer 9ifC,
>
> Thank you so much for your time, valuable comments and positive assessment of our work (good soundness, presentation and contribution). We are glad that you appreciate the clear motivation for evaluating agents on multi-stage time-series workflows, the broad and diverse dataset collection that enhances ecological validity, and the multi-artifact evaluation covering predictions, code quality, and trained models. We also appreciate the recognition of our agent-agnostic design, which enables flexible trajectory collection, as well as our documented process for adding new tasks, supporting scalability and long-term maintenance.
>
> > W1. The paper mixes the benchmark, the task generation mechanism, the multi artifact scoring, the trajectory data loop, and the time series focus without a clear primary to secondary hierarchy and without a concise contribution figure. It does not decompose difficulty across different time series task families and it does not design difficulty along agent reasoning dimensions.
>
> Thank you for the constructive feedback. In response, we clarify our contribution structure and introduce an explicit categorization of task difficulty.
>
> **Clarifying the Contribution Structure.** We clarify that our work is not a collection of disconnected components, but is guided by a single overarching question: **how should agents be evaluated in realistic ML engineering settings?** To address this, our benchmark is structured around three **complementary pillars** that jointly define our framework:
>
> - **Task Curation:** Expands the benchmarking scope through broader time-series task coverage from multiple real-world sources.
> - **ML Capability Coverage:** Covers both ML science and ML engineering tasks rather than only modeling.
> - **Evaluation Protocol:** Assesses multiple types of agent outcome artifacts and specific ML skills, combining objective and optional qualitative assessments.
>
> As summarized in Table 1 and Figure 1 in the original manuscript, each contribution maps directly to one of these pillars, which collectively implement our guiding principle. We will revise the manuscript to make this structure more explicit and streamline the contribution summary to improve clarity.
>
> **Difficulty Decomposition** As "difficulty" can be subjective, we instead categorize tasks using the structural complexity of the input data, which directly determines the reasoning capablities required for an agent to solve the tasks:
>
> - **Low:** Single CSV input. Requires standard file processing and schema understanding.
> - **Medium:** Inputs spanning multiple files or nested directories. Requires reasoning over file hierarchies and synthesizing information across different data structures.
> - **High:** Multiple files or directories from heterogeneous sources or modalities. Requires cross-modal reasoning in addition to handling multiple files/directories.
>
> The distribution of tasks across different families and difficulty levels is detailed below:
>
> Task Family | Low | Medium | High | Total |
> ------------|-----|--------|------|-------|
> Kaggle Challenges  |  5  |    7   |   1  |   13  |
> $\texttt{TimeSeriesGym}$ Originals |  4  |    8   |   2  |   14   |
> Derived Challenges |  4  |    0   |   2  |   6   |
> **Total** |  13  |    15   |   5  |   33   |
>
> We will incorporate this stratification into the revised manuscript to explicitly characterize the difficulty level distribution across task families.

---

> ### Author Response · Authors · 2025-11-26
> **Author Response (2/5)**
>
> > W2. Due to compute constraints most experiments are conducted on the Lite set of six tasks. Although the authors state that these tasks cover key skills there is no statistical validation of domain diversity or difficulty distribution which can bias conclusions toward low cost scenarios.
>
> Thank you for raising this important point. When constructing $\texttt{TimeSeriesGym-Lite}$, we intentionally select a diverse subset of tasks to preserve ML capability coverage. To further validate representativeness, we statistically compare domain coverage and difficulty distributions between $\texttt{TimeSeriesGym}$ and $\texttt{TimeSeriesGym-Lite}$.
>
> The domain distributions are detailed below. A Chi-square test with a p-value of 0.92 ($\chi^2 = 1.99$) shows no significant difference in domain distributions, indicating that **the Lite subset maintains the same domain diversity as the full benchmark.**
>
> Domain | $\texttt{TimeSeriesGym}$ | $\texttt{TimeSeriesGym-Lite}$ |
> -------|-------------|-------------|
> Healthcare      | 11 | 3 |
> Multi-domain    | 7  | 1 |
> Commerce & Finance | 6  | 1 |
> Weather      | 3 | 0 |
> Geology      | 2 | 0 |
> Housing      | 1 | 0 |
> Energy       | 1 | 0 |
>
> Similarly, we compare the task difficulty levels detailed below. A Chi-square test with a p-value of 0.95 ($\chi^2 = 0.09$) again shows **no significant difference in difficulty distribution.**
>
> Difficulty | $\texttt{TimeSeriesGym}$ | $\texttt{TimeSeriesGym-Lite}$ |
> -------|-------------|-------------|
> Low    | 13 | 2 |
> Medium | 15 | 3 |
> High   | 5  | 1 |
>
>
> Together, these tests validate that **$\texttt{TimeSeriesGym-Lite}$ is a statistically representative subset of $\texttt{TimeSeriesGym}$ in domain diversity and difficulty levels, enabling reliable evaluation under low-cost settings.** We will include these statistics and results in the revised manuscript to further clarify this representativeness.

---

> ### Author Response · Authors · 2025-11-26
> **Author Response (3/5)**
>
> > W3. The contribution is primarily engineering. The paper lacks a unifying methodological or theoretical insight. Although this is a benchmark paper it would benefit from a central methodological principle that guides design choices across sections.
>
> We believe that our paper is guided by a unifying methodological principle: **how to construct scalable benchmarks that evaluate agents in realistic ML engineering settings**, and our contributions and design choices across task curation, challenge design, and evaluation protocols are structured to support this goal. Section 3 articulates our framework, which formalizes how such agent benchmarks should be constructed. We will further emphasize this methodological principle in the revised manuscript to make it more explicit.
>
> ### Scalability
> We believe that the **scalability** of the benchmark is a core and novel contribution. Below we highlight numerous mechanisms that make `TimeSeriesGym` scalable.
>
> We believe that the following design choices make `TimeSeriesGym` scalable by design:
> 1. **Scaling skill-specific competitions:** We provide specialized tools to create skill-specific challenges (e.g., simulating missing data) and evaluate them. These skill simulators can be used with any number of "base" competations to generate a large number of diverse competitions which test for specific skills.
> 2. **Scaling agent outputs:** Our benchmarks provides tools to evaluate a large number of agent outputs, including prediction files, model artifacts, and code.
> 3. **Scaling agentic scaffolds:** Unlike existing benchmarks such as `MLGym`, `TimeSeriesGym` is agnostic to specific agent implementations. It is designed in a way that makes it easy to add new agent implemenatations.
> 4. **Data sources:** Kaggle comprises of an ever growing number of professional challenges, and many ML practitioners also have proprietary datasets on which they want to evaluate agents. Our benchmark is designed in a way to enable researchers to easily add and evaluate new tasks, whether Kaggle-style or not, as and when such challenges appear (see the following paragraph).
>
> Additionally, we believe that the following properties make `TimeSeriesGym` scalable in practice:
> 1. **Detailed documentation to add new challenges:** Our [detailed documentation](https://anonymous.4open.science/r/TimeSeriesGym-9CF6/documentation/) has allowed members of our community, but outside of our research team, to contribute a new challenge within 2 hours. Our publicly released anonymous code, had detailed step-by-step documentation to add new challenges and agentic scaffolds. While this evidence is anecdotal, nevertheless, we believe that it supports the scalability claim.
>
> We believe that these design decisions significantly lower the barrier and reduces the cost and implementation effort for reproducible agent evaluation, especially for practitioners who cannot release their data publicly.
>
>
> ### Novel Insights
> 1. **More time does not reliably improve agent performance.**
> Even with 2–3× more time (up to 12 hours / 150 steps), agents only produced reasonable submissions ~50% of the time, revealing diminishing returns from extended execution time.
>
> 2. **Agents fail to use time effectively or strategically.**
> Removing time reminders did not hurt performance—in fact, agents sometimes performed better—suggesting current agents do not manage time, reflect, or explore systematically.
>
> 3. **TimeSeriesGym reveals skill-specific weaknesses in agents.**
> Stratified analysis shows agents struggle with core ML engineering skills such as hyperparameter tuning and missing-data handling, while performing better on tasks like code migration depending on scaffold design.
>
> 4. **Benchmark exposes differential capabilities between scaffolds and LLMs.**
> AIDE outperforms alternatives on data-handling tasks, while others fail to produce valid submissions, surfacing meaningful skill gaps across agent frameworks.
>
> 5. **DeepSeek V3.2 demonstrates strong code translation but poor end-to-end ML engineering performance.**
> It excels at the ResNet migration task (8/9 tests passed) but fails on 5/6 tasks due to buggy code generation elsewhere, indicating highly uneven capability profiles.
>
> 6. **LLM-based feedback can hallucinate and mislead agent debugging.**
> Within the AIDE scaffold, erroneous code sometimes receives incorrect “successful” analyses, revealing a structural limitation of LLM-driven correction loops.
>
> 7. **TimeSeriesGym surfaces limitations not seen in Kaggle-style benchmarks.**
> Frameworks like AIDE—strong on standard Kaggle tasks—struggle on the more complex, flexible ML workflows required by TimeSeriesGym, validating the benchmark’s ability to reveal real-world ML engineering challenges.
>
>
> We hope we are able to convince you that our contributions far beyond "engineering". We believe that `TimeSeriesGym` is an integral step to build the next generation of effective and efficient LLM agent for ML engineering.

---

> ### Author Response · Authors · 2025-11-26
> **Author Response (4/5)**
>
> > W4. The table presents results for 4/50→12/150 and for Step wise reminder vs No reminder, but it does not include significance testing.
>
> Thank you for raising this point. Given that our evaluation uses three runs with identical random seeds across settings, we apply a **paired permutation test** comparing each variant to the default configuration (first row of each table). Results are shown below.
>
> ### Effect of Scaling Resources
>
> | Resources (hours/steps) | Valid Submission (%) |  Reasonable Submission (%) |
> |----|-----|----|
> |  4 / 50  | $66.7 \pm 16.7$ | $27.8 \pm 9.6$ |
> | 8 / 100  | $72.2 \pm 9.6^{(p = 1.0)}$  | $22.2 \pm 9.6^{(p = 1.0)}$ |
> | 12 / 150 | $61.1 \pm 9.6^{(p = .75)}$  | $50.0 \pm 0.0^{(p = .25)}$ |
>
>
> ### Effective Utilization of Time
>
> |  | Valid Submission (%) |  Reasonable Submission (%) |
> |----|-----|----|
> |  Step-wise reminder  | $66.7 \pm 16.7$ | $27.8 \pm 9.6$ |
> | No reminder  | $55.6 \pm 9.6^{(p = .5)}$  | $33.3 \pm 0.0^{(p = .5)}$ |
>
> Overall, the high p-values confirm that these differences are not statistically significant, supporting our findings that: (1) **agents do not improve with more time**, and (2) **agents do not utilize time effectively**.
>
>
> > W5. The definitions of Valid submission and Reasonable submission appear in prose in the methods section without a unified symbolization or an explicit decision boundary. This reduces clarity and hurts reproducibility.
>
>
> Thank you for pointing this out. We provide unified notation and explicit decision criteria for both metrics below.
>
> Let $S$ denote a submission, $G$ the grading function producing a numeric score (e.g., accuracy, MAE), and $\mathbb{I}(\cdot)$ the indicator function. We define two binary evaluation functions:
>
> $$ f_{\text{valid}} (S) \in \{0, 1 \}, \quad f_{\text{reasonable}} (S) \in \{0, 1 \} $$
>
> **Valid Submission** A submission is valid if the grader returns any non-null score:
>
> $$ f_{\text{valid}} (S) = \mathbb{I} (G(S) \neq \emptyset) $$
>
> **Reasonable Submission** The criterion is task-family specific.
>
> - **Kaggle competitions:** A submission is reasonable if it scores above the median on the competition’s public leaderboard. Let $P(G(S), L)$ denote the percentile of the score $G(S)$ relative to leaderboard $L$:
>
> $$ f_{\text{reasonable}} (S) = \mathbb{I} (P(G(S), L) > 0.5 ) $$
>
> - **$\texttt{TimeSeriesGym}$ original tasks:** A submission is reasonable if it includes a genuine modeling attempt. Let $M$ be a detector (regex-based or manual inspection) that outputs true when a modeling attempt is present:
>
> $$ f_{\text{reasonable}}(S) = \mathbb{I}(M(S) = \text{true}) $$
>
> We are adding these formal definitions in the revised manuscript to improve clarity and reproducibility.
>
> > Q1. Can you provide stratified results by task family with confidence intervals and tests for significance. This would help to understand which capabilities drive aggregate gains.
>
> Thank you for your suggestion. To enable more fine-grained comparison across tasks with different metric scales (e.g., RMSE vs. LogLoss), we introduce a novel metric: **normalized Percentile Score** ($1 - \frac{\text{rank}}{\text{total participants}}$), where 1.0 represents the first place on the leaderboard. For `TimeSeriesGym` original tasks, we establish "research leaderboards" that include the top 10 performances from the 100 most-cited papers for each dataset.
>
> The stratified results below indicate that performance gains primarily come from Kaggle challenges, which aligns with AIDE’s specific design for data science tasks. In contrast, `TimeSeriesGym` originals and derived tasks that require more flexible ML engineering skills (e.g., repository utilization, hyperparameter tuning) are challenging even for state-of-the-art agents.
>
> | Complexity | Num of Tasks | Normalized Percentile Score (Best@3) |
> | :--- | :---: | :---: |
> | **Kaggle Challenges** | 13 | $0.35 \pm 0.34$ |
> | **TimeSeriesGym Originals** | 14 | $0 \pm 0$ |
> | **Derived Challenges** | 6 | $0.05 \pm 0.07$ |
>
> We are adding these stratified results into the paper appendix to provide more fine-grained and diagnostic insights.
>
> > Q2. How do you ensure that the Lite subset is representative of the full benchmark. Please include quantitative evidence of coverage and difficulty distribution.
>
> Thank you for the question. As shown in our response to W2, we performed Chi-square tests comparing **domain coverage** and **difficulty distributions** between $\texttt{TimeSeriesGym}$ and $\texttt{TimeSeriesGym-Lite}$. The resulting p-values (0.92 and 0.95) indicate **no statistically significant difference** along either axis.
>
> Therefore, **$\texttt{TimeSeriesGym-Lite}$ is a statistically representative subset of $\texttt{TimeSeriesGym}$ in terms of task families, domain coverage, and difficulty levels.** Please refer to our response to W2 for full statistical details.

---

> > ### Author Response · Authors · 2025-11-26
> > **Author Response (5/5)**
> >
> > > Q3. Can you formalize the validity and reasonableness criteria using symbols and thresholds and show calibration plots or decision curves.
> >
> > Thank you for the question. We provide a unified symbolic definition of both validity and reasonableness in our response to W5, including explicit decision thresholds. These thresholds are **fixed and deterministic** (median rank on leaderboard for Kaggle tasks, and presence of genuine modeling attempt for $\texttt{TimeSeriesGym}$ originals). As the criteria do not involve tunable thresholds, they do not yield calibration curves, ROC curves, or decision trade-offs by design.
> >
> > > Q4. What steps can be taken to reduce reliance on leaderboard medians or external competitions that may change over time.
> >
> > Thank you for the question. All competitions included in $\texttt{TimeSeriesGym}$ are archived, and their public leaderboards are fixed. To ensure reproducibility independent of external sources, we additionally store a **static snapshot** of the leaderboard scores of each competition in the benchmark.

---

### Official Review · Reviewer_CCeW · 2025-11-01

**Soundness:** 3
**Presentation:** 2
**Contribution:** 2
**Rating:** 4
**Confidence:** 4

**Summary:**

This paper proposes a benchmark framework named TimeSeriesGym, designed to evaluate the capabilities of AI agents in time series machine learning engineering tasks. The core idea is to provide a scalable and agent-agnostic evaluation environment that encompasses time series challenges across multiple domains, while assessing various outputs generated by agents—including prediction files, code, and models.

**Strengths:**

1、 The benchmark collects and designs tasks based on real-world data science scenarios, including Kaggle competition problems and practical research tasks such as code migration and model evaluation. These challenges span a wide range of skills—including data processing, model construction, and code understanding and adaptation—reflecting the multifaceted challenges faced by real-world machine learning engineers.

2、 TimeSeriesGym evaluates multiple forms of agent outputs, not only focusing on prediction accuracy and error metrics, but also assessing code generation, model artifacts, and more. Additionally, it introduces optional LLM-based review mechanisms (e.g., code auditing) to supplement quality evaluation.

3、 The authors provide tooling mechanisms that enable large-scale automatic generation of new tasks, making the benchmark highly extensible and sustainable, with the ability to continuously incorporate new challenges.

**Weaknesses:**

1、 The primary contribution of the paper lies in the construction of the benchmark. Many of its ideas—such as using LLMs for code review, incorporating multi-source tasks, and adopting multi-metric evaluations—are extensions and integrations of existing work rather than entirely novel innovations.

2、 Although the paper lists several existing benchmarks, the distinctions and connections between TimeSeriesGym and those benchmarks are not sufficiently elaborated. For example, beyond the domain difference, how does TimeSeriesGym improve upon the evaluation philosophy of MLE-Bench? A deeper comparative analysis would strengthen the positioning of the proposed framework.

3、 Among the 33 current challenges, tasks sourced from Kaggle tend to be well-structured and may have standard solutions, while the original TimeSeriesGym tasks are often highly complex. GPT-4 achieves only a 12.1% “reasonable solution” rate on TimeSeriesGym, with many tasks yielding no valid outputs. It is recommended that future task sets include medium-difficulty challenges to ensure a smoother difficulty gradient, enabling better tracking of agent performance from beginner to advanced levels. Additionally, establishing baselines for each task—such as simple algorithms or human-level performance—would help characterize task difficulty and the potential improvement space for agents.

**Questions:**

1、 As the authors noted, due to funding and time constraints, most experiments were conducted on a Lite subset of 6 tasks. This raises two concerns: (A) Can the Lite subset sufficiently represent the full benchmark? Although the authors selected diverse tasks, the sample size of six remains relatively small. (B)Some results—such as the extension of agent steps to 12 hours—were only tested on the Lite subset. It is unclear whether similar trends would hold across the full benchmark.

2、 Additionally, at line 76 below Figure 1, there appears to be white-colored text revealing the authors' institutional affiliation (“Ⓒ 2025 Auton Lab, Carnegie Mellon University”). This may violate the double-blind review policy and should be addressed.

---

> ### Author Response · Authors · 2025-11-26
> **Author Response (1/3)**
>
> Dear Reviewer CCew,
>
> Thank you so much for your time and valuable comments. We are pleased that you recognize our use of real-world data science scenarios, our evaluation of diverse agent outputs—including code and model artifacts with optional LLM-based auditing—and our tooling for large-scale automatic task generation that ensures the benchmark’s continued extensibility.
>
> > W1. The primary contribution of the paper lies in the construction of the benchmark. Many of its ideas—such as using LLMs for code review, incorporating multi-source tasks, and adopting multi-metric evaluations—are extensions and integrations of existing work rather than entirely novel innovations.
>
> `TimeSeriesGym` is a indeed a **scalable** benchmark to evaluate LLM agents on real-world ML engineering problems, spanning a wide range of skills, and multiple forms of outputs. In addition to the benchmark, which we believe is a valuable and novel contribution by itself, `TimeSeriesGym` also provides practical insights to motivate future work.
>
> We believe that the **scalability** of the benchmark is a core and novel contribution. Below we highlight numerous mechanisms that make `TimeSeriesGym` scalable.
>
> We believe that the following design choices make `TimeSeriesGym` scalable by design:
> 1. **Scaling skill-specific competitions:** We provide specialized tools to create skill-specific challenges (e.g., simulating missing data) and evaluate them. These skill simulators can be used with any number of "base" competations to generate a large number of diverse competitions which test for specific skills.
> 2. **Scaling agent outputs:** Our benchmarks provides tools to evaluate a large number of agent outputs, including prediction files, model artifacts, and code.
> 3. **Scaling agentic scaffolds:** Unlike existing benchmarks such as `MLGym`, `TimeSeriesGym` is agnostic to specific agent implementations. It is designed in a way that makes it easy to add new agent implemenatations.
> 4. **Data sources:** Kaggle comprises of an ever growing number of professional challenges, and many ML practitioners also have proprietary datasets on which they want to evaluate agents. Our benchmark is designed in a way to enable researchers to easily add and evaluate new tasks, whether Kaggle-style or not, as and when such challenges appear (see the following paragraph).
>
> Additionally, we believe that the following properties make `TimeSeriesGym` scalable in practice:
> 1. **Detailed documentation to add new challenges:** Our [detailed documentation](https://anonymous.4open.science/r/TimeSeriesGym-9CF6/documentation/) has allowed members of our community, but outside of our research team, to contribute a new challenge within 2 hours. Our publicly released anonymous code, had detailed step-by-step documentation to add new challenges and agentic scaffolds. While this evidence is anecdotal, nevertheless, we believe that it supports the scalability claim.
>
> We believe that these design decisions significantly lower the barrier and reduces the cost and implementation effort for reproducible agent evaluation, especially for practitioners who cannot release their data publicly.
>
> In summary, our primary contribution is a **scalable benchmark that formulates ML engineering skills as explicit evaluable targets and isolates these skills from an end-to-end ML workflow**. While prior work has explored code review or multi-source inputs separately, to our knowledge no existing benchmark scales along all of the dimensions discussed above as `TimeSeriesGym`.
>
> We are revising the manuscript to highlight these distinctions more clearly.

---

> ### Author Response · Authors · 2025-11-26
> **Author Response (2/3)**
>
> > W2. Although the paper lists several existing benchmarks, the distinctions and connections between TimeSeriesGym and those benchmarks are not sufficiently elaborated. For example, beyond the domain difference, how does TimeSeriesGym improve upon the evaluation philosophy of MLE-Bench? A deeper comparative analysis would strengthen the positioning of the proposed framework.
>
>
> We agree that we can improve the presentation of the significant differences between `TimeSeriesGym` and other existing benchmarks (including `MLE-Bench`). We are actively working on revising our paper to make the following differences clearer.
>
> `MLE-Bench` is a well-designed benchmark, which primarily evaluates LLM agents on their ability to solve text and image Kaggle challenges. While `TimeSeriesGym` also sources challenges from Kaggle, it is designed to evaluate LLM agents on their ability to do ML engineering, of which solving Kaggle challenges is a small (but important) part. `TimeSeriesGym` has several important distinctions from `MLE-Bench`:
>
> 1. **Coverage of Time Series Tasks:** As shown in Table 1, `TimeSeriesGym` has far more challenges that require modeling time series data compared to any ML engineering benchmark. Time series is an important and underrepresented modality. We show that many real-world time series problems (e.g., PTB-XL ECG classification) can be reformulated into a fully automated, agentic evaluation pipeline, which is not demonstrated by prior benchmarks.
> 2. **Original challenges:** `MLE-Bench` exclusively sources challenges from Kaggle. Our extensive (and growing) documentation makes it easier for the community to add new Kaggle-based challenges easier. But Kaggle challenges alone do not reflect real-world (time series) machine learning engineering problems, such as hyperparameter search, repository utilization, and API integration capabilities that existing benchmarks, including `MLE-Bench`, do not isolate or measure directly. `TimeSeriesGym` includes original challenges, which represent realistic ML engineering problems motivated by years of experience of the research team.
> 3. **Broader ML capability coverage:** `TimeSeriesGym` has skill-simulators (e.g., data missingness handling, hyperparameter search) that none of the other benchmarks have, which enable researchers to evaluate specific capabilities of LLM agents.
> 4. **Signficantly Harder:** `TimeSeriesGym` is challenging for state-of-the-art LLMs and agentic scaffolds. Meanwhile, existing benchmarks such as `MLE-Bench` are much easier for these agents. Note that, at the time of writing the paper, `o1-preview` with the AIDE scaffold achieved medal in 16.9% of the challenges. Meanwhile, our updated results reveal that the default setup (AIDE + GPT-4o) achieved a percentile score > 0.5 (above median) in only 3 out of 13 Kaggle competitions, which reveal a significant gap in the time series modeling and ML engineering capabilites of state-of-the-art agents.
> 5. **Multimodal outputs:** `MLE-Bench` can only evaluate CSV prediction files, whereas `TimeSeriesGym` can also evaluate model artifacts and code. These modalities are common in ML engineering practice.
> 6. **Holistic evaluation:** Our evaluation methodology deliberately combines multiple assessment approaches: quantitative metrics (e.g., accuracy, mean absolute error), programmatic analysis (e.g., regular expression matching, code inspection), and optional qualitative evaluation (LLM-as-a-judge). Each challenge in `TimeSeriesGym` is evaluated using quantitative metrics and _optionally_ subjective metrics (LLM judge).
>
> In the original paper, Table 1 provides a succint and visual comparison of `TimeSeriesGym` with existing benchmarks (including `MLE-Bench`) along three key dimensions: **Task Source, ML Capability Coverage,** and **Evaluation Protocol**. Below, we summarize how $\texttt{TimeSeriesGym}$ complements and improves upon $\texttt{MLE-Bench}$ along these three axes:
>
> | | $\texttt{MLE-Bench}$ |  $\texttt{TimeSeriesGym}$ |
> |---|---|---|
> | Task Source | Only Kaggle | Combination of **Kaggle, GitHub repositories, and hand-crafted tasks** |
> | ML Capability | Focuses primarily on ML science tasks (e.g., modeling) | Covers both **ML science and engineering** (e.g., repository utilization) tasks  |
> | Evaluation Protocol | Evaluates only CSV outputs using objective metrics | Evaluates **multiple artifacts** (CSV outputs, code, model artifacts) and **specific ML skills** (e.g., data handling) from a **holistic perspective** (both objective metrics and qualitative code assessment)  |

---

> > ### Author Response · Authors · 2025-11-26
> > **Author Response (3/3)**
> >
> > > W3. Among the 33 current challenges, tasks sourced from Kaggle tend to be well-structured and may have standard solutions, while the original TimeSeriesGym tasks are often highly complex. GPT-4 achieves only a 12.1% “reasonable solution” rate on TimeSeriesGym, with many tasks yielding no valid outputs. It is recommended that future task sets include medium-difficulty challenges to ensure a smoother difficulty gradient, enabling better tracking of agent performance from beginner to advanced levels. Additionally, establishing baselines for each task—such as simple algorithms or human-level performance—would help characterize task difficulty and the potential improvement space for agents.
> >
> > Thank you for the valuable suggestion. As task "difficulty" can be subjective, we instead categorize challenges using the structural complexity of the input data, which directly determines the level of reasoning required for an agent:
> >
> > - **Low:** Single CSV input. Requires standard file processing and schema understanding.
> > - **Medium:** Inputs spanning multiple files or nested directories. Requires reasoning over file hierarchies and synthesizing information across different data structures.
> > - **High:** Multiple files or directories from heterogeneous sources or modalities. Requires cross-modal reasoning in addition to handling multiple files/directories.
> >
> > To enable more fine-grained comparison across tasks with different metric scales (e.g., RMSE vs. LogLoss), we also introduce a novel metric: **normalized Percentile Score** ($1 - \frac{\text{rank}}{\text{total participants}}$), where 1.0 represents the first place on the leaderboard. For `TimeSeriesGym` original tasks, we establish "research leaderboards" that include the top 10 performances from the 100 most-cited papers for each dataset.
> >
> > The table below illustrates the performance of the default agent (AIDE + GPT-4o), indicating substantial headroom across all complexity levels, especially for high-complexity tasks requiring multi-source or multi-modality reasoning.
> >
> > | Complexity | Num of Tasks | Average Normalized Percentile Score (Best@3) |
> > | :--- | :---: | :---: |
> > | **Low** | 13 | 0.174 |
> > | **Medium** | 15 | 0.162 |
> > | **High** | 5 | 0.035 |
> >
> > We will include the difficulty categorization and Research Leaderboards in future public releases (and revision of the paper) to further strengthen the practical utility of our benchmark.
> >
> > > Q1. As the authors noted, due to funding and time constraints, most experiments were conducted on a Lite subset of 6 tasks. This raises two concerns: (A) Can the Lite subset sufficiently represent the full benchmark? Although the authors selected diverse tasks, the sample size of six remains relatively small. (B)Some results—such as the extension of agent steps to 12 hours—were only tested on the Lite subset. It is unclear whether similar trends would hold across the full benchmark.
> >
> > Thank you for the question. When constructing $\texttt{TimeSeriesGym-Lite}$, we intentionally select a diverse subset of tasks to preserve ML capability coverage. To further validate the representativeness, we statistically compare the domain coverage and difficulty distributions between $\texttt{TimeSeriesGym}$ and $\texttt{TimeSeriesGym-Lite}$.
> >
> > The domain distributions are detailed below. A Chi-square test with a p-value of 0.92 ($\chi^2 = 1.99$) shows no significant difference in domain distributions, indicating that **the Lite subset maintains the same domain diversity as the full benchmark.**
> >
> > Domain | $\texttt{TimeSeriesGym}$ | $\texttt{TimeSeriesGym-Lite}$ |
> > -------|-------------|-------------|
> > Healthcare      | 11 | 3 |
> > Multi-domain    | 7  | 1 |
> > Commerce & Finance | 6  | 1 |
> > Weather      | 3 | 0 |
> > Geology      | 2 | 0 |
> > Housing      | 1 | 0 |
> > Energy       | 1 | 0 |
> >
> > Similarly, we compare the task difficulty levels detailed below. A Chi-square test with a p-value of 0.95 ($\chi^2 = 0.09$) again shows **no significant difference in difficulty distribution.**
> >
> > Difficulty | $\texttt{TimeSeriesGym}$ | $\texttt{TimeSeriesGym-Lite}$ |
> > -------|-------------|-------------|
> > Low    | 13 | 2 |
> > Medium | 15 | 3 |
> > High   | 5  | 1 |
> >
> >
> > Together, these tests validate that **$\texttt{TimeSeriesGym-Lite}$ is a statistically representative subset of $\texttt{TimeSeriesGym}$ in domain diversity and difficulty levels, enabling reliable evaluation under low-cost settings.** We will include these statistics and results in the revised manuscript to further clarify this representativeness.

---

### Official Review · Reviewer_c71P · 2025-11-01

**Soundness:** 3
**Presentation:** 2
**Contribution:** 2
**Rating:** 4
**Confidence:** 5

**Summary:**

The authors introduce TimeSeriesGym, a new benchmarking framework designed to evaluate AI agents on machine learning (ML) engineering tasks specifically for the time series domain. The paper argues that existing benchmarks are flawed, focusing too narrowly on well-defined Kaggle-style problems, lacking scalability, and only evaluating final submission files. TimeSeriesGym aims to solve this by sourcing diverse challenges (including 20 "Original" tasks based on real-world engineering like code migration and hyperparameter tuning), designing a framework to grade multiple artifacts (code, models, and submissions), and employing a holistic evaluation that includes both quantitative metrics and qualitative "LLM-as-a-judge" assessments. The framework also claims to be scalable with tools for new challenge creation. The paper's main experimental finding is that current state-of-the-art agents perform very poorly on these tasks, often failing to produce even a "reasonable" submission.

**Strengths:**

Identifies a Clear and Important Gap: The paper is correct that time series is an underserved domain in agentic benchmarking. Creating a dedicated benchmark for this is a valuable contribution.

Focuses on "ML Engineering," Not Just Modeling: The strongest part of the paper is its inclusion of "TimeSeriesGym Originals" (Table 4). Challenges like "Convert ResNet TensorFlow implementation to PyTorch" or "Improve PTB-XL ECG Classification Code" are excellent, real-world tasks that go beyond the standard "train-a-model-on-a-CSV" format.

Holistic Evaluation Concept: The ambition to evaluate multiple artifacts (code, models) and use a "two-faceted grading approach" (quantitative and qualitative, see Appendix E) is the right direction for a comprehensive benchmark.

Scalability as a Design Goal: Designing the benchmark to be scalable from the start (even if not fully proven) is a smart, forward-thinking approach that addresses a major flaw in static benchmarks.

**Weaknesses:**

**Critically Flawed Evaluation of Kaggle Challenges:**

The paper's primary evaluation metrics, "Valid Submission (%)" and "Reasonable Submission (%)," are insufficient.

- The benchmark fails to report the actual leaderboard scores or ranks for the 13 included Kaggle competitions. This is a significant omission, as these are the most standardized, competitive tasks in the dataset.

- The bar for a "Reasonable Submission" is set at scoring "above median on the competition's public leaderboard" (Section 4). This threshold is exceptionally low and provides no real insight into an agent's capabilities. An agent that barely surpasses the median is treated identically to one that achieves a state-of-the-art, gold-medal-winning score.

- This lack of granular, comparable results (as seen in Table 10) makes it impossible for the research community to evaluate an agent's true performance or benchmark progress against established human or SOTA baselines on these problems.

**Conceded Risk of Data Leakage:**
The paper's reliance on popular, public Kaggle competitions fundamentally compromises its ability to evaluate modern LLM-based agents. The authors' defense—that agents "performed poorly anyway" or that the benchmark is "scalable" to add new tasks—is unconvincing. It effectively concedes that the current Kaggle-based portion of the benchmark is unsuitable for reliably evaluating frontier models, as performance may be confounded by memorization.

Subjective and Non-Scalable Evaluation Protocol:

- The paper's claim of being a "scalable benchmark" is directly undermined by its own evaluation methodology for the "Originals" challenges.

- For non-Kaggle tasks, "reasonableness" is determined by the authors "manually inspecting" if a "genuine modeling attempt" was made (Section 4, Metrics). A benchmark that requires subjective, manual intervention from its creators for a primary metric is, by definition, not scalable or objective.

- The proposal to use "LLM-as-a-judge" (Appendix E) as a secondary evaluation method introduces a notoriously unreliable, biased, and difficult-to-reproduce component, which is not a substitute for rigorous, objective, quantitative metrics.

- The authors themselves admit the protocol's flaws in Section 5 ("Defining and measuring success"), stating that "our current evaluation approaches have inherent limitations."

**Miscalibrated Difficulty (Floor Effects):**

- The reported results are so poor that the benchmark, in its current form, largely fails as a diagnostic tool.

- The best agent on the full benchmark (AIDE + GPT-4.1) only achieved a "reasonable" submission 12.1% of the time (Section 4.1).

- Even on the hand-picked, "easy" Lite benchmark, the best agent (AIDE + Claude 3.7) only succeeded 38.9% of the time (Table 2).

These results indicate a significant floor effect. The tasks are currently too difficult for SOTA agents, preventing any meaningful differentiation between models or scaffolds. The benchmark primarily demonstrates that all agents fail, offering little insight into why they fail or which approaches are more promising.

**Questions:**

See the weakness.

---

> ### Author Response · Authors · 2025-11-25
> **Author Response (1/2)**
>
> Dear Reviewer c71P,
>
> Thank you so much for your valuable comments and time. Below are our detailed responses to your concerns:
>
> ### **W1. Critically Flawed Evaluation of Kaggle Challenges**
>
> We appreciate the reviewer's suggestion.
>
> > The benchmark fails to report the actual leaderboard scores or ranks for the 13 included Kaggle competitions. This is a significant omission, as these are the most standardized, competitive tasks in the dataset.
>
> Please note that most Kaggle competitions have a hidden, private test set. The training and testing sets in most benchmarks, including `TimeSeriesGym` are constructed from the public datasets. Therefore, directly comparing our agents to scores on the leaderboard is incorrect and can be misleading. However, we do agree with the reviewer, that raw scores (e.g., RMSE vs. LogLoss) can be difficult to compare across tasks, and there should be a way for us to compare our agents with existing leaderboards.
>
> Therefore, to address the reviewer's concern we include a novel metric: **normalized Percentile Score** ($1 - \frac{\text{rank}}{\text{total participants}}$), where 1.0 represents the first place on the leaderboard. While imperfect, this score is more robust than leaderboard ranks, and allows agentic scaffolds to be compared across tasks.
>
> Our updated results reveal that the default setup (AIDE + GPT-4o) achieved a percentile score > 0.5 (above median) in only 3 out of 13 Kaggle competitions.
>
> **This empirical evidence suggests that the "Reasonable" threshold (above median) is in fact not "exceptionally low"**, but rather a significant barrier for current state-of-the-art agents. We use this binary metric not to obscure performance, but because current capability gaps make granular distinctions at the top (e.g., Gold vs. Silver) mathematically irrelevant until agents can consistently surpass the median.
>
> Unlike existing benchmarks which have started showing signs of saturation with agents achieve gold and silver medals, `TimeSeriesGym` was designed to be a challenging, and a real test for an agent's ability to do ML engineering. As mentioned in the paper, we view `TimeSeriesGym` as a living benchmark - as agents do better, we will, along with the community come up with new evaluation metrics (e.g., medals when they become necessary) and novel, harder challenges.
>
>
> ###  **W2. Conceded Risk of Data Leakage**
>
> Thank you for raising this important concern. We acknowledge that public Kaggle competitions may overlap with portions of an LLM’s pretraining corpus, and we agree that this risk must be carefully examined.
>
> Beyond simply noting the poor empirical performance, we conducted a formal familiarity analysis to directly assess whether frontier LLMs show signs of prior exposure or memorization risk to these tasks. As detailed in Lines 364–372 and Appendix C.4, we computed familiarity scores based on GPT-4.1’s mean token probabilities on task descriptions, and compared the score distribution against that of `MLE-Bench`. A Kolmogorov–Smirnov (KS) test on these two distributions resulted in $p = 0.363$, indicating no statistically significant difference. **This suggests that the tasks used in our benchmark **do not appear unusually “familiar” or memorization-prone** relative to other widely used evaluation benchmarks.**

---

> ### Author Response · Authors · 2025-11-25
> **Author Response (2/2)**
>
> ###  **W3. Subjective and Non-Scalable Evaluation Protocol**
>
> `TimeSeriesGym` is a scalable benchmark because it supports a variety of agentic scaffolds, new challenges can be added with relative ease, and it supports a variety of evaluation methods.
>
> We agree with the reviewer that: (1) manual evaluation limits the scalability of the evaluation, and (2) LLM judges can be unreliable. However, we use manual inspection to evaluate whether the agent made a reasonable attempt, whereas LLM judges are exclusively used for code reviews.
>
> However, philosophically speaking: manual evaluation complements the unreliability of LLM judges, whereas LLM judges enable scalability. The inclusion of both these methodologies enables `TimeSeriesGym` to be both reliable, and scalable, to a certain extent.
>
> That said, manually verification is _not_ time consuming. `TimeSeriesGym` provides an easy to read journal of agent trajectories, which provides an easy and quick way for reseachers to decide if an agent has made a genuine attempt. For our research team, it took a few seconds per competition.
>
> However, to enable scalable reproduction of the study, the public release will include a Regex-based verification script that automatically checks for genuine modeling attempts. While rule-based checking generally has limitations (e.g., syntactic rigidity), our approach is robust in practice as we control the Docker environment and know exactly which packages are pre-installed. Although agents can technically install external libraries, our heuristic explicitly covers the vast majority of standard ML frameworks. We view this as a deliberate trade-off that avoids the stochasticity of LLM judges while ensuring reproducibility at scale, and we will continue to refine the verification rules to ensure its consistency with human judgements.
>
> We explicitly emphasize that LLM-as-a-judge is **NOT** part of our standard evaluation protocol, but only an _optional_ qualitative analysis. We share the reviewer's concern regarding its unreliability, which is exactly why **all our standard evaluations rely solely on objective scoring functions** (as shown in Table 10). For instance, the specific PTB-XL task mentioned in Appendix E utilizes a fully deterministic scoring function based on classification accuracy, and the LLM-based judgement was solely for qualitative insights but not for scoring.
>
> ### **W4. Miscalibrated Difficulty (Floor Effects)**
>
> We agree that several tasks in the benchmark are challenging. However, we respectfully disagree that low absolute performance renders the benchmark uninformative. Instead, we argue that these results do provide meaningful diagnostic insight into current agentic capabilities.
>
> First, even when overall success rates are low, the benchmark exposes systematic failure modes across agents. As detailed in Appendix D, we observe several recurring patterns:
>
> **Long-context and multi-turn failures.** Agents often drop or overwrite critical information, such as ignoring the provided data-loading interface and reverting to incorrect defaults.
>
> **Scaffold-specific weaknesses.** For example, AIDE’s single-file generation structure makes repository-level tasks inherently difficult, revealing limits of this design choice.
>
> **Hallucinations under high task complexity.** When tasks require navigating a multi-file codebase and identifying subtle dependencies, models often fall back to fabricated outputs rather than reasoning through the repository.
>
> These are precisely the types of insights a diagnostic benchmark should reveal: where and how current scaffolds break under realistic conditions involving long contexts, external code, and multi-step workflows.
>
> Philosophically speaking, our goal was not to design a benchmark where the compared agents "do well". Instead, we priorized including realistic and challenging ML engineering problems, which can inspire the next generation of agentic scaffolds.
>
> _We appreciate your thoughtful feedback and have carefully reflected on your reviews, conducted further experiments, and we are also making substantial revisions to the paper. If our responses have addressed your concerns, we would appreciate if you would consider revising your evaluation. Thank you again for your time and careful review!_
>
> Best regards, \
> Authors of TimeSeriesGym

---

### Author Response · Authors · 2025-12-02
**General Response (1/2)**

Dear Reviewers and Area Chair,

We really appreciate your time and effort in reviewing our paper! Across reviews, several consistent strengths emerged.

1. **Fills an important gap** (Reviewer c71P, E1hF): Reviewers agreed that TimeSeriesGym fills an important and previously unaddressed gap in benchmarking real-world time-series ML engineering pipelines, going beyond traditional forecasting or single-step prediction tasks.
2. **Real-world ML engineering tasks** (Reviewer c71P, CCeW, 9ifC): Reviewers highlighted the realism and breadth of the benchmark’s task suite, noting that it spans diverse domains and includes practical engineering challenges—such as data processing, code migration, model evaluation, and other multi-stage workflows—that better reflect how ML engineers work in practice.
3. **Holistic, multi-artifact evaluation framework** (Reviewer c71P, CCeW, 9ifC, E1hF): Reviewers emphasized the value of our holistic, multi-artifact evaluation framework, which integrates quantitative metrics with qualitative assessments (e.g., code audits) and evaluates outputs ranging from predictions to executable code and trained models.
4. **`TimeSeriesGym` is scalable by design** (Reviewer c71P, CCeW, 9ifC): Multiple reviewers recognized scalability as a core design principle, supported by detailed documentation, tooling for automated task generation, and mechanisms that enable the benchmark to evolve over time.
5. **Agent agnostic design** (Reviewer 9ifC): Finally, a reviewer appreciated the agent-agnostic nature of the benchmark, which supports diverse frameworks and allows trajectory collection for future training and research.
6. **Novel insights** (Response to Reviewer 9ifC): While responding to Reviewer 9ifC, we summarized several novel empirical insights, which further reinforce `TimeSeriesGym`'s value:
    - More time does not reliably improve agent performance-- even with 2–3× more time, agents produce reasonable outputs only ~50% of the time.
    - Agents fail to use time strategically—removing time reminders sometimes improves performance.
    - Skill-specific weaknesses emerge (e.g., hyperparameter tuning, missing-data handling), while scaffolds succeed on easier tasks like code migration.
    - Scaffolds differ sharply in capability—AIDE outperforms alternatives on data-handling tasks, but all scaffolds struggle with more complex workflows.
    - LLM-based feedback loops can hallucinate, misleading debugging processes.
    - `TimeSeriesGym` exposes limitations invisible in Kaggle-style benchmarks, validating that time-series ML engineering requires skills beyond standard leaderboard optimization.

At the same time, we received valuable feedback from our reviewers. Their feedback highlighted critical areas for refinement, and we have conducted a comprehensive suite of new experiments, analyses, and modifications to the paper, to address these concerns during the rebuttal phase.

1. **Open-source (like frontier LLMs) struggle as ML engineering agents.** Reviewer E1hF requested broader model coverage beyond closed-source frontier models. During the rebuttal, we evaluated *DeepSeek V3.2 (Reasoner + Chat)* using the AIDE scaffold on `TimeSeriesGym-Lite`. Results show that existing open-source agents are far from capable ML engineers, and that `TimeSeriesGym` is well-suited to reveal such capability gaps quickly.
2. **`TimeSeriesGym-Lite` is a representative subset of the full benchmark.** Reviewers CCeW and 9ifC asked whether the 6-task Lite version preserves the diversity of the full benchmark. To validate this, we statistically compared domain and difficulty distributions of the lite and full version of the benchmark using Chi-square tests. These analyses confirm that `TimeSeriesGym-Lite` is a **statistically representative** subset of `TimeSeriesGym`, enabling meaningful and low-cost evaluation for research groups without large compute budgets.

---

> ### Author Response · Authors · 2025-12-03
> **General Response (2/2)**
>
> 3. **TimeSeriesGym brings a step-change compared to existing ML engineering benchmarks.** Reviewers CCeW and E1hF noted the importance of positioning our benchmark relative to prior work. We clarified that MLE-Bench is strong for text/image Kaggle tasks, **TimeSeriesGym uniquely targets real-world ML engineering problems**, with several distinctions:
>     * **Broader coverage of time series tasks**, an underserved modality in ML engineering benchmarks.
>     * **Original challenges** (e.g., hyper-parameter search, missing-data handling, API integration) not captured by Kaggle-only benchmarks.
>     * **Skill simulators** enabling capability-specific analysis (a feature absent from existing benchmarks).
>     * **Multimodal outputs** (predictions, code, models) and **holistic evaluation** beyond CSV submissions.
>     * **Substantially higher difficulty**, with SOTA agents achieving median-level performance on only 3 of 13 Kaggle tasks.
>       These differences support the reviewers' observation that `TimeSeriesGym` fills a unique and important gap.
>
> 4. **TimeSeriesGym is scalable by design.** Reviewer E1hF wanted stronger justification for our benchmark's scalability. We emphasized that scalability arises from **structural design**, not anecdotal evidence:
>     * Programmatic task-generation tools for skill-specific variants.
>     * Detailed documentation and templates for adding new challenges.
>     * Evaluation logic that is **agent-agnostic and framework-agnostic**, enabling longevity as agents evolve.
>     * Support for diverse agent artifacts (code, models, predictions).
>
> 5. **Robust and scalable agent evaluation.** In response to Reviewer c71P and 9ifC's questions on cross-task comparability, we introduce a novel **Normalized Percentile Score** that allows agents to be compared consistently across challenges with heterogeneous metrics. During the rebuttal, we also improved the robustness of our evaluations by developing a **regex and LLM-judge based system to _automatically_ identify genuine modeling attempts**. These **mechanisms produce a scalable, reproducible evaluation framework aligned with reviewers' praise for holistic grading**.
>
>
> 6. **Insights from performance stratified by difficulty and challenge type.**
>    Reviewers CCeW and 9ifc asked how challenge difficulty is determined and whether it correlates with agent performance. We categorize tasks by **input structural complexity** (low/medium/high) and analyze agent scores under this lens. Our results revealed **substantial capability gaps**, especially in tasks requiring multi-file reasoning or multimodal synthesis. We will include these categorizations and “Research Leaderboards” in the revised paper.
>
> 7. **Significance testing for reviewer concerns:** As requested by Reviewer 9ifc , we conducted paired permutation tests comparing each variant (increased time; no-reminder setting) to the default configuration. The **high p-values confirm no statistically significant improvements**, strengthening our claims that (1) more time does not help, and (2) agents do not use time effectively.
>
> Changes to our paper made as a result of your feedback are shown in deep blue.
>
> Taken together, these revisions directly and comprehensively address every weakness raised by our reviewers. We expanded model coverage, validated the representativeness of the Lite subset, clarified the benchmark’s positioning, strengthened the scalability argument, improved the robustness and comparability of evaluations, and added new analyses on difficulty, task structure, and statistical significance. Importantly, many of the originally perceived weaknesses have now become **demonstrated strengths** of the benchmark: `TimeSeriesGym` now features broader model evaluations, validated task design, principled scalability, and a significantly more rigorous evaluation pipeline. We believe these changes have materially improved the clarity, completeness, and scientific rigor of the paper, and we hope that you all agree that the benchmark is now substantially stronger as a result of your thoughtful feedback.

---

### Meta-Review · Area_Chair_sVBR · 2026-01-02

**Summary:**

The paper proposes a benchmark called "TimeSeriesGym", which is positioned as a scalable benchmark for time series ML agents in similar spirit to MLE Bench. Overall, all reviewers acknowledged the necessity for such benchmarking, pointing out that the important domain of time series data remains understudied in agentic AI. At the same time, all reviewers agree that there are fundamental limitations to the proposed evaluation methodology, that the benchmark dataset composition is in parts unclear, and that more work needs to be done to substantiate scalability, enhance result interpretation, and rigorously clarify the benchmark's constructions. The AC agrees with these points, and as pointed out below, does not believe these points to have been addressed sufficiently in the rebuttal and revision. The AC thus recommends to reject the paper in its present form.

**Reviewer Concerns:**

All reviewers had a set of shared concerns. These concerns could roughly be split into three aspects. 1. Differentiation from MLE-Bench beyond "engineering" an "MLE-Bench"-like benchmark for time series, 2. The construction of the benchmark regarding a lack of difficulty gradient and very low performance being non-indicative of agent utility and lacking informativeness regarding downstream improvements, 3. Evaluation choices regarding questionable design practices, such as the (auxiliary) use of LLM-as-a-judge, leading to concerns over how the benchmark will be scaled down the line.

Going through the rebuttal and revisions, a larger set of clarifications has been added to differentiate TimeSeriesGym from MLE-Bench and point out its necessity. Whereas the inspiration is evident, the AC believes this point has been addressed sufficiently. However, regarding the other two points, which compose the unanimous shared concerns, it is less clear that the rebuttal has resolved these aspects in a clear and precise manner. Many of the rebuttal responses are explanatory in nature, and the revisions seem to either provide clarifications or include additional material that seems somewhat tangential to the raised concerns. For instance, there has been a large concern over the hard difficulty of the benchmark, the lack of taxonomy regarding difficulty, and the subsequent lack of informativeness of the largely failing agents (who at large do not have the necessary capabilities to solve the benchmark yet). Here, the rebuttal has added a valuable categorization and additional experimentation regarding input complexity, yet the AC fails to see how this addition addresses the actual concern. Although valuable, it does not address the main concerns. The same can be said about the third point, where clarifications have largely been provided and acknowledgements written to point out understood limitations, but no actionable path has been proposed for future scaling of the benchmark that would do better. As such, the AC unfortunately believes the biggest concerns to be fully outstanding.

**Reviewer Scores:**

Initial reviewer scores were all leaning towards rejection. Only one reviewer seems to have been able to participate in the discussion prior to the freeze. This reviewer did not seem convinced by the lengthy rebuttal. Overall, the AC would like to point out that it is naturally hard to extrapolate potential scores based on a non-existent discussion. In an attempt to nevertheless do so, the AC believes that the other reviewers would have reacted similarly and would have at best changed their score only slightly. The rationale for this belief is provided in above box already, but can be succinctly summarized as the rebuttal being largely tangential to the main concerns pointed out. The AC believes it is unlikely the reviewers would have changed their scores towards unanimous acceptance, as the revisions largely address superficial improvements and provide auxiliary content, leaving the main critique untouched. As such, the AC thinks at least two of the reviewers would have retained their score, given that the deeper questions wrt benchmark construction and rigorous evaluation remain unresolved.

---

### Decision · Program_Chairs · 2026-01-26

Reject